

# Global form of flavor symmetry groups
# in $4d$ $\mathcal{N} = 2$ theories of class S

**Lakshya Bhardwaj**

Mathematical Institute, University of Oxford, Andrew Wiles Building,
Woodstock Road, Oxford, OX2 6GG, UK

## Abstract

We provide a systematic method to deduce the global form of flavor symmetry groups in $4d$ $\mathcal{N} = 2$ theories obtained by compactifying $6d$ $\mathcal{N} = (2, 0)$ superconformal field theories (SCFTs) on a Riemann surface carrying regular punctures and possibly outer-automorphism twist lines. Apriori, this method only determines the group associated to the manifest part of the flavor symmetry algebra, but often this information is enough to determine the group associated to the full enhanced flavor symmetry algebra. Such cases include some interesting and well-studied $4d$ $\mathcal{N} = 2$ SCFTs like the Minahan-Nemeschansky theories. The symmetry groups obtained via this method match with the symmetry groups obtained using a Lagrangian description if such a description arises in some duality frame. Moreover, we check that the proposed symmetry groups are consistent with the superconformal indices available in the literature. As another application, our method finds distinct global forms of flavor symmetry group for pairs of interacting $4d$ $\mathcal{N} = 2$ SCFTs (recently pointed out in the literature) whose Coulomb branch dimensions, flavor algebras and levels coincide (along with other invariants), but nonetheless are distinct SCFTs.

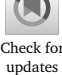

# 1 Introduction

Traditionally, when studying a continuous flavor (0-form [1]) symmetry of a quantum field theory (QFT) $\mathfrak{T}$, one studies only the Lie algebra $\mathfrak{f}$ associated to the symmetry. However, recent studies have shown that a lot of important data about the QFT $\mathfrak{T}$ is encoded in its flavor symmetry *group*, which is a Lie group $\mathcal{F}$ whose associated Lie algebra is the flavor symmetry algebra $\mathfrak{f}$ [1]. For example:

- $\mathcal{F}$ determines the space of possible background bundles for flavor symmetry that $\mathfrak{T}$ can be coupled to. These backgrounds allow the characteristic classes of $\mathcal{F}$ to be turned on, but a characteristic class of some other global form $\mathcal{F}'$ of $\mathfrak{f}$ which is not a characteristic class of $\mathcal{F}$ is not allowed to be turned on.

- The knowledge of $\mathcal{F}$ is crucial to the determination of the set of possible discrete 't Hooft anomalies involving the 0-form flavor symmetry of $\mathfrak{T}$. This is because the characteristic classes of $\mathcal{F}$ can enter into various discrete 't Hooft anomalies, and different global forms of $\mathfrak{f}$ have different characteristic classes. The presence of such discrete 't Hooft anomalies in $\mathfrak{T}$ can be revealed by observing phase anomaly in the correlation functions of $\mathfrak{T}$ in the presence of a background for $\mathcal{F}$ that turns on characteristic classes associated to $\mathcal{F}$.

- The knowledge of $\mathcal{F}$ is also crucial to the determination of 2-group and higher-group symmetries of $\mathfrak{T}$ involving the 0-form flavor symmetry group. Some higher-group symmetries can be understood as deformations of the space of possible backgrounds for some other $p$-form symmetry of $\mathfrak{T}$ in the presence of non-trivial backgrounds of $\mathcal{F}$, with the precise form of deformation being captured by characteristic classes of $\mathcal{F}$. Thus different forms of $\mathcal{F}$ for a fixed $\mathfrak{f}$ allow different possible higher-group symmetry structures for $\mathfrak{T}$.

- All the local operators in $\mathfrak{T}$ must transform in those representations of $\mathfrak{f}$ that are also allowed representations of $\mathcal{F}$. [2]

In this paper, we provide a method to determine the flavor symmetry group[3] $\mathcal{F}$ of an arbitrary 4d $\mathcal{N} = 2$ theory obtained by compactifying a 6d $\mathcal{N} = (2,0)$ SCFT on a Riemann surface of arbitrary genus $g$, carrying arbitrary regular punctures which can be both twisted and untwisted, and arbitrary closed outer-automorphism twist lines. These 4d $\mathcal{N} = 2$ theories form a subclass of the 4d $\mathcal{N} = 2$ theories of Class S [3]. More general Class S theories can be obtained by including irregular punctures, which we do not address in this work.

We begin in Section 2.1 by reviewing the definition of flavor symmetry group in terms of the allowed classes of representations of local operators in the theory. We also discuss that

---

[1]Often, one also says that $\mathcal{F}$ is the 'global form' of the flavor symmetry $\mathfrak{f}$.

[2]Thus knowledge of $\mathcal{F}$ can potentially be used to constrain and simplify conformal bootstrap approaches to study $\mathfrak{T}$.

[3]See also a recent paper [2] determining the global forms of flavor symmetry groups of 5d SCFTs using their M-theory constructions.

this definition can be understood as encoding the space of background flavor bundles that the theory can be coupled to. We shift to the discussion of flavor symmetry groups of gauge theories in Section 2.2, where the matter content can include generalized matter in the form of interacting conformal theories. We emphasize that the flavor symmetry group captures the representations of the gauge-invariant local operators. We also discuss a Pontryagin dual way of defining the flavor symmetry group of gauge theories, which provides an alternate point of view for the flavor symmetry group as being consisted of those elements that act non-trivially on the matter content and cannot be undone by a gauge transformation. In Section 2.3, we consider situations in which one knows the flavor symmetry group associated to a subalgebra of the full flavor symmetry algebra. We discuss the constraints imposed on the full flavor symmetry group from the knowledge of the flavor symmetry group associated to the subalgebra. Such a situation occurs frequently in the study of flavor symmetries of $4d$ $\mathcal{N} = 2$ Class S theories.

In Section 3, we discuss manifest flavor symmetry groups in $4d$ $\mathcal{N} = 2$ theories of Class S containing arbitrary untwisted and twisted regular punctures and arbitrary closed twist lines. A key role in the analysis is played by the surface defects of the $6d$ $(2,0)$ theory. We review some properties of these defects and discuss their circle compactification in Section 3.1, focusing on the correspondence between surface defects in the $6d$ $(2,0)$ theory and gauge Wilson line defects in the $5d$ $\mathcal{N} = 2$ SYM theory resulting from circle compactification. In Section 3.2, we view regular punctures in Class S construction as boundary conditions in $5d$ $\mathcal{N} = 2$ SYM theory, which allows us to determine a set of local operators contributed to the $4d$ Class S theory 'locally' by each puncture. In Section 3.3, we discuss additional local operators in the $4d$ Class S theory that arise from wrapping surface defects of the $6d$ theory along the Riemann surface, which can be thought of as the 'global' contribution to the set of local operators in the $4d$ theory. We propose that the local operators discussed in Sections 3.2 and 3.3 account for the flavor center charges of all the local operators in the $4d$ Class S theory. Then, the manifest flavor symmetry group is obtained by applying the analysis of Section 2.1.

Section 4 illustrates the procedure of Section 3 with a variety of examples. The examples have been chosen such that there is an alternative method for verifying the manifest flavor symmetry group. The examples appearing in Section 4.1 are free hypers transforming in various representations of the manifest flavor symmetry algebra, and hence one can deduce their manifest flavor symmetry groups in two ways: using the method of Section 3 and the method of Section 2.2 with trivial gauge group. The examples appearing in Section 4.2 admit a duality frame where the $4d$ $\mathcal{N} = 2$ theory becomes a weakly coupled $4d$ $\mathcal{N} = 2$ gauge theory and hence one can again deduce their manifest flavor symmetry groups in two ways: using the method of Section 3 and the method of Section 2.2. The examples appearing in Section 4.3 are strongly coupled interacting $4d$ $\mathcal{N} = 2$ SCFTs where the local operator representations deduced using the analysis of Section 3 are checked against the representations appearing in superconformal indices of these SCFTs available in the literature. The examples discussed in this section include $E_6, E_7$ Minahan-Nemeschansky theories, the $T_N$ trinion theories and the recently discussed $\widetilde{T}_3$ trinion theory [4] which arises from less well-studied twisted $A_2$ compactifications.

In Section 4.3, we also illustrate how the manifest flavor symmetry group is often enough to determine uniquely the full, true flavor symmetry group. In addition to this, in this section we also consider a pair of interacting $4d$ $\mathcal{N} = 2$ SCFTs whose Coulomb branch dimensions, Weyl anomaly coefficients, flavor symmetry algebras and levels coincide, but nonetheless they are two distinct SCFTs. Such examples were recently pointed out in [5] and it was also proposed there that the two SCFTs have distinct global forms of flavor symmetry groups by studying the Schur index. Applying our method to the pair, we find the same flavor symmetry groups as proposed in [5].

Beyond these examples, the methods of this paper can be used to deduce manifest, and in many cases true, flavor symmetry groups of arbitrary $4d$ $\mathcal{N} = 2$ Class S theories with regular punctures and outer-automorphism twists. It would be an interesting future direction to incorporate irregular punctures into this analysis.

## 2 Generalities about Flavor Symmetry Groups

### 2.1 Definition of Flavor Symmetry Group

Consider a theory $\mathfrak{T}$ with a flavor symmetry algebra $\mathfrak{f}$. The local operators of $\mathfrak{T}$ transform in various representations of $\mathfrak{f}$. We want to study the representations of $\mathfrak{f}$ that are or aren't carried by local operators of $\mathfrak{T}$. First of all, independent of the specifics of $\mathfrak{T}$, we have a current operator associated to the flavor symmetry in $\mathfrak{T}$ which transforms in the adjoint representation of $\mathfrak{f}$. Taking the OPE of the current with itself, we can generate local operators transforming in all representations of $\mathfrak{f}$ that can be generated by taking tensor products of the adjoint representation with itself.

Now depending on $\mathfrak{T}$, we might have local operators carrying other representations of $\mathfrak{f}$. To characterize such representations, let us decompose $\mathfrak{f} = \mathfrak{f}_{na} \oplus \mathfrak{u}(1)^a$ such that $\mathfrak{f}_{na}$ is a non-abelian semi-simple Lie algebra and $\mathfrak{u}(1)^a$ is the abelian part of $\mathfrak{f}$. Now let us define $F = F_{na} \times U(1)^a$ to be a group whose associated Lie algebra is $\mathfrak{f}$, such that $F_{na}$ is the simply connected group associated to $\mathfrak{f}_{na}$ and $U(1)^a$ is a group whose associated Lie algebra is $\mathfrak{u}(1)^a$. Moreover, let $Z_F$ denote the center of $F$, which can be written as $Z_F = Z_{F,na} \times U(1)^a$ where $Z_{F,na}$ is the center of the simply connected group $F_{na}$.

A theory-specific local operator $\mathcal{O}$ can then be associated to an element $\alpha_{\mathcal{O}}$ of the Pontryagin dual $\widehat{Z}_F$ of $Z_F$. This element $\alpha_{\mathcal{O}}$ captures the charge of the representation $R$ of $\mathcal{O}$ under the center $Z_F$ of $F$. The elements of $\widehat{Z}_F$ associated to all the local operators in $\mathfrak{T}$ define a subgroup $Y_F$ of the abelian group $\widehat{Z}_F$ due to the following reasons:

- If a local operator $\mathcal{O}_1$ is associated to an element $\alpha_{\mathcal{O}_1} \in \widehat{Z}_F$ and a local operator $\mathcal{O}_2$ is associated to an element $\alpha_{\mathcal{O}_2} \in \widehat{Z}_F$, then by taking the OPE of $\mathcal{O}_1$ and $\mathcal{O}_2$ we can find a local operator associated to the element[4] $\alpha_1 + \alpha_2 \in \widehat{Z}_F$.

- If a local operator $\mathcal{O}$ is associated to an element $\alpha \in \widehat{Z}_F$, then the CPT conjugate local operator $\mathcal{O}^*$ is associated to the element[5] $-\alpha \in \widehat{Z}_F$.

- The identity local operator and the current operator are associated to the element $0 \in \widehat{Z}_F$ since they have a trivial charge under the center $Z_F$ of $F$.

Now, let us define

$$\widehat{\mathcal{Z}} = \widehat{Z}_F / Y_F \,, \tag{1}$$

which comes with a projection map

$$\widehat{Z}_F \to \widehat{\mathcal{Z}} \,. \tag{2}$$

Taking the Pontryagin duals, we obtain an injection

$$\mathcal{Z} \to Z_F \,, \tag{3}$$

which identifies $\mathcal{Z}$ as a subgroup of $Z_F$. The *flavor symmetry group* $\mathcal{F}$ of the theory is then

$$\mathcal{F} = F/\mathcal{Z} \,. \tag{4}$$

---

[4]Addition denotes the group law of $\widehat{Z}_F$.

[5]Here $-\alpha$ denotes the inverse of $\alpha$ in $\widehat{Z}_F$.

Thus, the flavor symmetry group $\mathcal{F}$ of a theory $\mathfrak{T}$ with flavor symmetry algebra $\mathfrak{f}$ is defined to be a group with the following properties:

- The Lie algebra associated to $\mathcal{F}$ is $\mathfrak{f}$.

- All the representations of $\mathfrak{f}$ formed by local operators of $\mathfrak{T}$ are allowed representations of $\mathcal{F}$.

- Any allowed representation of $\mathcal{F}$ is realized by some local operator of $\mathfrak{T}$.

The relevance of this definition can be understood by noticing that a theory $\mathfrak{T}$ with flavor symmetry group $\mathcal{F}$ can be coupled to a background flavor bundle for $\mathcal{F}$. A local operator $\mathcal{O}$ transforming in a representation $R$ of $\mathcal{F}$ lives at the end of a flavor Wilson line in representation $R$. If, instead, we try to couple $\mathfrak{T}$ to a background gauge bundle for some other global form $\mathcal{F}'$ of $\mathfrak{f}$ which does not allow a representation $R$ appearing in $\mathfrak{T}$, then the corresponding local operator $\mathcal{O}$ cannot be inserted as there is no corresponding flavor Wilson line available. On the other hand, we can couple $\mathfrak{T}$ to a background gauge bundle for a global form $\mathcal{F}''$ of $\mathfrak{f}$ which allows all the representations appearing in $\mathcal{T}$ but also allows some representations not appearing in $\mathfrak{T}$, but such $\mathcal{F}''$ bundles are all special cases of $\mathcal{F}$ bundles.

## 2.2 Flavor Symmetry Groups of Gauge Theories

Consider a gauge theory with gauge group $G$ whose center is $Z_G$. Let the flavor symmetry associated to the matter content of the theory form a flavor algebra $\mathfrak{f} = \mathfrak{f}_{na} \oplus \mathfrak{u}(1)^a$ as in the previous subsection. Let us also define $F = F_{na} \times U(1)^a$ with center $Z_F$ as in the previous subsection.

The charges of matter content[6] of the gauge theory under $Z_G \times Z_F$ generate a sub-lattice $\mathcal{M} \subseteq \widehat{Z}_G \times \widehat{Z}_F$. This sub-lattice $\mathcal{M}$ captures the representations of local operators in the theory. That is, consider a representation $R_G$ of $G$ having charge $\alpha_G \in \widehat{Z}_G$ and a representation $R_F$ of $F$ having charge $\alpha_F \in \widehat{Z}_F$, such that the element $(\alpha_G, \alpha_F) \in \mathcal{M}$. Then there exists a local operator $\mathcal{O}$ in the theory that transforms in representation $R_G \otimes R_F$ of $G \times F$ and lives at the end of a gauge Wilson line defect carrying the representation $R_G$ of $G$ [7].

The group $Y_F$ defined in the previous subsection is obtained as

$$Y_F = \mathcal{M} \cap \widehat{Z}_F. \tag{5}$$

That is, $Y_F$ is the subgroup of $\mathcal{M}$ formed by elements of the form $(0, *) \in \widehat{Z}_G \times \widehat{Z}_F$. Then $Y_F$ captures the charges under $Z_F$ of *gauge-invariant* local operators of the theory[8]. The flavor symmetry group $\mathcal{F}$ of the gauge theory is then given by (4).

One point to note is that $Y_F$ and hence $\mathcal{F}$ depend only on the gauge algebra $\mathfrak{g}$ and not on the precise global form of $\mathfrak{g}$ used. That is, if we use some other global form $G'$ of $\mathfrak{g}$ to perform the above computation, then we obtain the same $Y_F$. This is because changing $G$ to $G'$ scales the gauge charges $\alpha_G$, but the matter content contributing to $Y_F$ has $\alpha_G = 0$ and hence $Y_F$ is left invariant by the scaling.

---

[6]This matter content may be standard perturbative matter or include generalized matter in the form of conformal theories whose (part of) flavor symmetry is gauged by $G$. In the latter case of generalized matter, the "charges of matter content" refers to the charges of genuine local operators in the conformal theory describing generalized matter. In any case, in Section 4, we will use the contents of this section with perturbative matter only.

[7]If $R_G$ is non-trivial, then such a local operator is often called a "non-genuine" local operator of the theory. For the purposes of this paper, a non-genuine local operator $\mathcal{O}$ is one that is constrained to live at the end of a non-trivial line defect $\mathcal{L}$, and cannot exist independently without the presence of $\mathcal{L}$. In the previous subsection, and in the rest of this paper, we use the term 'local operator' for a genuine local operator, unless otherwise clear from the context.

[8]Gauge invariant local operators in a gauge theory are genuine local operators, as they do not need to be inserted at the end of a non-trivial gauge Wilson line to be well-defined.

An equivalent way of computing these groups is as follows. Let $\mathcal{S}$ denote the full structure group of the gauge theory that acts non-trivially on the matter content, which can be written as

$$\mathcal{S} = \frac{G \times F}{\mathcal{E}}, \tag{6}$$

where $\mathcal{E} \subseteq Z_G \times Z_F$ under which the matter content does not transform. We then obtain $\mathcal{Z}$ as

$$\mathcal{Z} = \pi_F(\mathcal{E}), \tag{7}$$

where $\pi_F(\mathcal{E}) \subseteq Z_F$ is the image of $\mathcal{E}$ under the projection map

$$\pi_F : Z_G \times Z_F \to Z_F, \tag{8}$$

which projects onto the second factor of $Z_G \times Z_F$. Stated more physically, the group $\mathcal{Z}$ can be identified as the subgroup of $Z_F$ whose elements either act trivially on the matter content or act like an element of $Z_G$ (and hence an element of the gauge group $G$). The elements of the former type do not act on the theory, while the elements of the latter type are part of the gauge group and hence should not be regarded as genuine flavor symmetries of the theory. Thus, the flavor symmetry group $\mathcal{F}$ of the gauge theory should be $\mathcal{F} = F/\mathcal{Z}$, which indeed coincides with the general definition (4).

## 2.3 Relationship Between Manifest and True Flavor Symmetry Groups

In this paper, we study flavor symmetry groups of $4d\ \mathcal{N} = 2$ Class S theories which are produced by compactifying $6d\ \mathcal{N} = (2,0)$ theories on a Riemann surface with punctures. In such a compactification, a part of the flavor symmetry algebra $\mathfrak{f}_{\mathcal{P}_i} \subseteq \mathfrak{f}$ is associated to each puncture $\mathcal{P}_i$. Thus, the manifest flavor symmetry is $\mathfrak{f}_m = \bigoplus_i \mathfrak{f}_{\mathcal{P}_i} \subseteq \mathfrak{f}$, but it is often the case that the manifest flavor symmetry $\mathfrak{f}_m$ is a proper subalgebra of the true flavor symmetry algebra $\mathfrak{f}$, i.e. $\mathfrak{f}_m \subset \mathfrak{f}$. In such a situation, the methods presented in this paper can be used to study only the flavor group associated to $\mathfrak{f}_m$, which constrains the flavor group associated to the full flavor symmetry $\mathfrak{f}$ but might not uniquely fix it. See the discussion below for more details.

We can apply the analysis of the previous two subsections to obtain the manifest flavor symmetry group $\mathcal{F}_m$ associated to the manifest flavor symmetry algebra $\mathfrak{f}_m$. The group $\mathcal{F}_m$ is related to the true flavor symmetry group $\mathcal{F}$ as follows:

- The manifest flavor symmetry group $\mathcal{F}_m$ is a subgroup of the true flavor symmetry group $\mathcal{F}$.

- If some other global form $\mathcal{F}'_m$ of $\mathfrak{f}_m$ is also a subgroup of $\mathcal{F}$, then we have

$$\mathcal{F}_m = \mathcal{F}'_m / \mathcal{Z}'_m, \tag{9}$$

  where $\mathcal{Z}'_m$ is a subgroup of the center of the group $\mathcal{F}'_m$.

In other words, $\mathcal{F}_m$ is the "minimal" global form of $\mathfrak{f}_m$ which is a subgroup of $\mathcal{F}$. The manifest flavor symmetry group $\mathcal{F}_m$ captures the allowed background bundles that one can turn on for the manifest part of flavor symmetry.

The above two conditions can also be viewed as constraining the possible forms of the true flavor symmetry group $\mathcal{F}$ using the knowledge of $\mathcal{F}_m$. That is, let $\mathcal{F}_\alpha$ be the various global forms of $\mathfrak{f}$ such that $\mathcal{F}_m \subseteq \mathcal{F}_\alpha$ for each $\alpha$, and if $\mathcal{F}'_m$ is some other global form of $\mathfrak{f}_m$ satisfying $\mathcal{F}'_m \subseteq \mathcal{F}_\alpha$ then $\mathcal{F}'_m$ is such that $\mathcal{F}_m = \mathcal{F}'_m / \mathcal{Z}'_m$ with $\mathcal{Z}'_m$ being a subgroup of the center of $\mathcal{F}'_m$. Then, the knowledge of $\mathcal{F}_m$ allows us to deduce that the true flavor symmetry group $\mathcal{F}$ must be one of the groups $\mathcal{F}_\alpha$. In case there is a single choice of $\mathcal{F}_\alpha$ satisfying the above two conditions, then we can determine that the true flavor symmetry group must be $\mathcal{F} = \mathcal{F}_\alpha$.

For practical use, let us phrase the above two conditions in terms of representations:

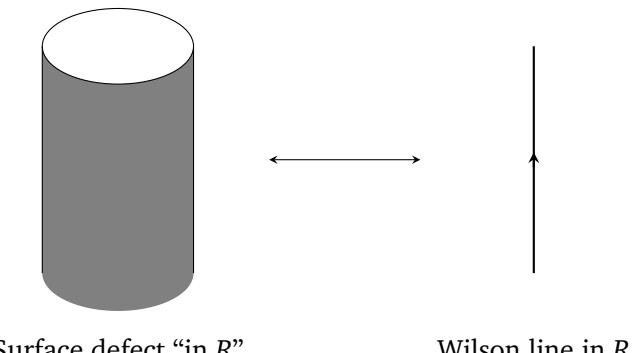

Surface defect "in $R$"  Wilson line in $R$

Figure 1: Compactifying a surface defect of $6d$ $(2,0)$ theory of type $\mathfrak{g}$ along $S^1$ direction in spacetime leads to a gauge Wilson line defect in $5d$ $\mathcal{N}=2$ $\mathfrak{g}$ SYM. Equivalently, a gauge Wilson line in the $5d$ theory lifts to a circle compactified surface defect of the $6d$ theory. If the Wilson line transforms in representation $R$ of $\mathfrak{g}$, then we label the the corresponding surface defect by the representation $R$.

- Take an irrep $R$ of $\mathcal{F}$ and decompose it as $R = \bigoplus_i R_i$ where $R_i$ are irreps of $\mathfrak{f}_m$. If each $R_i$ for each $R$ is also an irrep of $\mathcal{F}_m$, then $\mathcal{F}_m \subseteq \mathcal{F}$.

- Consider all the irreps $R_i$ of $\mathfrak{f}_m$ descending from all irreps $R$ of $\mathcal{F}$. These irreps $R_i$ form a sublattice $Y_{F_m} \subseteq \widehat{Z}_{F_m}$ where $Z_{F_m}$ is the center of the group $F_m$ associated to $\mathfrak{f}_m$ as explained in Section 2.1. Then, $\mathcal{F}_m$ is the global form of $\mathfrak{f}_m$ determined by (4) (with input group $F$ taken to be $F_m$).

## 3 Flavor Symmetry Groups in Class S

### 3.1 Surface Defects in $6d$ $(2,0)$ SCFTs and Their Circle Compactification

A $6d$ $\mathcal{N}=(2,0)$ SCFT is specified by a finite A, D, E Lie algebra $\mathfrak{g}$. Consider such a $6d$ SCFT and compactify it on a circle of non-zero size. The resulting theory is $5d$ $\mathcal{N}=2$ pure super Yang-Mills (SYM) theory with gauge algebra[9] $\mathfrak{g}$. The gauge Wilson lines of the $5d$ theory arise from the circle compactification of dimension-2 surface operators of the $6d$ theory. Consequently, we can characterize the surface defects of the $6d$ theory in terms of representations of $\mathfrak{g}$. See Figure 1. Similarly, local operators screening[10] gauge Wilson lines of the $5d$ theory can be understood as arising from circle compactification of line defects[11] screening the dimension-2 surface defects of the $6d$ theory. See Figure 2. The gauge Wilson line defects of $5d$ $\mathcal{N}=2$ SYM with gauge algebra $\mathfrak{g}$ are characterized, modulo screenings, by elements of $\widehat{Z}(\mathcal{G})$ which is the Pontryagin dual of the center $Z(\mathcal{G})$ of the simply connected group $\mathcal{G}$ associated to the gauge algebra $\mathfrak{g}$ [12]. Consequently, the dimension-2 surface defects of a $6d$ $\mathcal{N}=(2,0)$ SCFT of type $\mathfrak{g}$ can be characterized, modulo screenings, by elements of $\widehat{Z}(\mathcal{G})$.

A $6d$ $\mathcal{N}=(2,0)$ SCFT of type $\mathfrak{g}$ admits a discrete 0-form symmetry group $\mathcal{O}_{\mathfrak{g}}$ formed by outer-automorphisms (modulo inner automorphisms) of $\mathfrak{g}$. See Table 1. These outer-

---

[9]Notice that we are not specifying the gauge group. Thus the $5d$ theory has mutually non-local operators and it is an example of what is known as a relative QFT.

[10]A line defect $\mathcal{L}$ is "screened" in a theory $\mathfrak{T}$ if there exists a non-genuine local operator $\mathcal{O}$ in $\mathfrak{T}$ that lives at the end of $\mathcal{L}$.

[11]Akin to above, these line defects are non-genuine in the sense that they are constrained to live at the ends of surface defects.

[12]As we will discuss later, this statement is modified for $5d$ $\mathcal{N}=2$ $\mathfrak{sp}(n)$ SYM theory with discrete theta angle $\pi$. But, since in our current context, $\mathfrak{g}$ is of A, D, E type, this modification does not concern us at the moment.

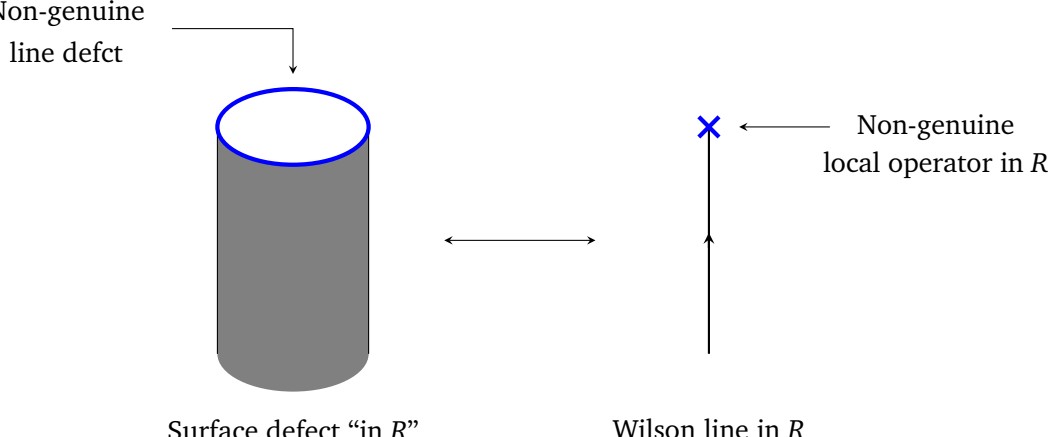

Non-genuine line defct

Non-genuine local operator in $R$

Surface defect "in $R$"

Wilson line in $R$

Figure 2: A surface defect ending at a non-genuine line defect in the $6d$ theory leads to, upon circle compactification, to a gauge Wilson line defect ending at a non-gauge-invariant local operator in the $5d$ theory. Equivalently, a gauge Wilson line defect ending at a non-gauge-invariant local operator in the $5d$ theory lifts in the $6d$ theory to a surface defect ending at a non-genuine line defect compactified along the circle.

automorphisms act on the nodes of the Dynkin diagram of $\mathfrak{g}$ as follows:

$\mathfrak{su}(2n)$, $\langle o \rangle = \mathbb{Z}_2$:

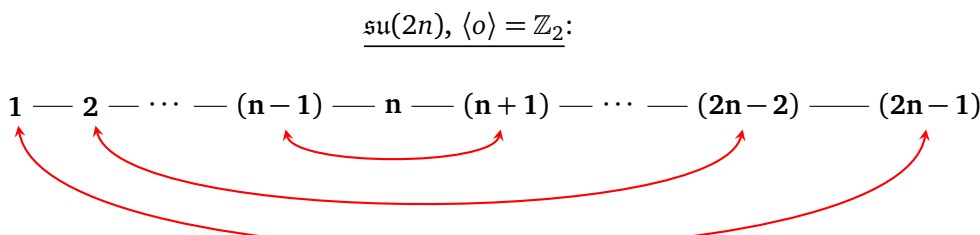

$\mathfrak{su}(2n+1)$, $\langle o \rangle = \mathbb{Z}_2$:

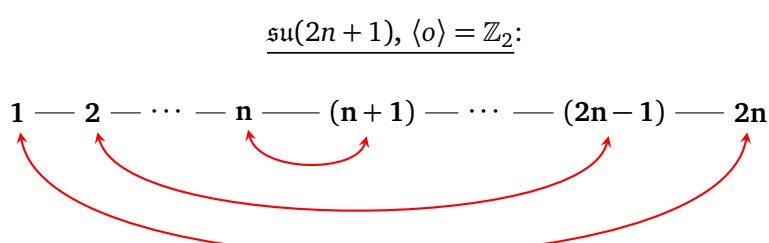

$\mathfrak{so}(2n)$, $n \geq 5$, $\langle o \rangle = \mathbb{Z}_2$:

$$1 \text{ --- } 2 \text{ --- } 3 \text{ --- } \cdots \text{ --- } (n-3) \text{ ---- } (n-2) \overset{n}{\underset{(n-1)}{<}}$$

$\mathfrak{so}(8)$, $\langle o \rangle = \mathbb{Z}_2^v$:    $\mathfrak{so}(8)$, $\langle o \rangle = \mathbb{Z}_2^s$:    $\mathfrak{so}(8)$, $\langle o \rangle = \mathbb{Z}_2^c$:    $\mathfrak{so}(8)$, $\langle o \rangle = \mathbb{Z}_3$:

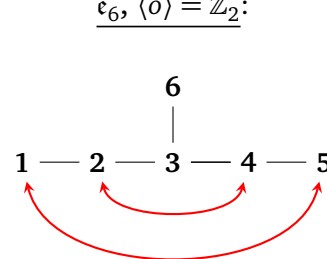

$\mathfrak{e}_6$, $\langle o \rangle = \mathbb{Z}_2$:

When we compactify the theory on a circle, we can turn on a non-trivial holonomy for the background gauge field associated to $\mathcal{O}_{\mathfrak{g}}$ along the circle. This is often referred to as compactifying the 6$d$ theory with an outer-automorphism "twist" along the circle. The holonomy is specified by an element $o \in \mathcal{O}_{\mathfrak{g}}$ to which is associated a subalgebra[13] $\mathfrak{h}_o$ of $\mathfrak{g}$ left invariant by the action of $o$.

Upon circle compactificatication with an $o$-twist, we obtain 5$d$ $\mathcal{N} = 2$ pure SYM theory with gauge algebra $\mathfrak{h}_o^\vee$ which is the Langlands dual of $\mathfrak{h}_o$. The action of Langlands duality on the Dynkin diagram is such the direction of edges is reversed. This action is non-trivial on the following algebras [14] :

$\mathfrak{sp}(n)$:

$1 \longrightarrow 2 \longrightarrow \cdots \longrightarrow (n-1) \Longleftarrow n \qquad \longleftrightarrow \qquad$

$\mathfrak{so}(2n+1)$:

$1 \longrightarrow 2 \longrightarrow \cdots \longrightarrow (n-1) \Longrightarrow n$

$\mathfrak{f}_4$:

$1 \longrightarrow 2 \Longleftarrow 3 \longrightarrow 4 \qquad \longleftrightarrow \qquad$

$\mathfrak{f}_4$:

$1 \longrightarrow 2 \Longrightarrow 3 \longrightarrow 4$

$\mathfrak{g}_2$:

$1 \Lleftarrow 2 \qquad \longleftrightarrow \qquad$

$\mathfrak{g}_2$:

$1 \Rrightarrow 2$

In all other cases, the action of Langlands duality is trivial.

Gauge Wilson line defects of the 5$d$ theory descend by wrapping surface defects of 6$d$ theory along the circle. A surface defect of the 6$d$ theory characterized by representation $R$ of

---

[13]We actually pick a particular outer-automorphism in the class $o$ of outer-automorphisms and $\mathfrak{h}_o$ is the subalgebra left invariant by this particular outer-automorphism.

[14]Notice that the action can be non-trivial even if the initial and final algebras are the same.

$\mathfrak{g}$ can be wrapped along the circle only if $R$ is left invariant by the action of $o$. This is ensured if the highest weights of the representation are left invariant by $o$, which means the following. Let $w$ be a highest weights and let $w_i$ be its Dynkin coefficients where $i$ labels the nodes in the Dynkin diagram of $\mathfrak{g}$. Now $o$ acts on node $i$ and sends it to another node $o(i)$. $R$ is left invariant by the action of $o$ if we have $w_i = w_{o(i)}$ for all of its highest weights $w$.

Now suppose $R$ is a representation of $\mathfrak{g}$ left invariant by $o$, then it descends to a representation $R_o$ of $\mathfrak{h}_o$ as follows. For each highest weight $w$ of $R$, we have a highest weight $w'$ of $R_o$ such that $w'_{i'} = w_i$ where $i'$ is a node in the Dynkin diagram of $\mathfrak{h}_o$ and $i$ is a node in the Dynkin diagram of $\mathfrak{g}$ which projects to $i'$ under the identification of Dynkin nodes induced by the action of $o$ on Dynkin nodes of $\mathfrak{g}$.

Now wrap a surface defect of the $6d$ theory characterized by a representation $R$ of $\mathfrak{g}$ left invariant by $o$. The resulting gauge Wilson line in the $5d$ theory is characterized by representation $R_o^\vee$ of $\mathfrak{h}_o^\vee$ whose highest weights have the same Dynkin coefficients as the highest weights of the representation $R_o$ of $\mathfrak{h}_o$, but now these highest weights are associated to the nodes in the Dynkin diagram of the Langlands dual algebra $\mathfrak{h}_o^\vee$ according to the action of Langlands duality displayed above.

Let us discuss this relationship between gauge Wilson lines of $5d$ $\mathfrak{h}_o^\vee$ theory and the surface defects of the $6d$ $\mathfrak{g}$ theory in more detail for every case:

- Compactifying $6d$ $\mathfrak{g} = \mathfrak{su}(2n)$ theory with $\mathbb{Z}_2$ outer-automorphism twist results in $5d$ $\mathfrak{h}_o^\vee = \mathfrak{so}(2n+1)$ gauge theory. The gauge Wilson lines of the $5d$ theory can be generated by taking products of the gauge Wilson line in the spinor irrep of $\mathfrak{so}(2n+1)$ gauge algebra. This gauge Wilson line is produced by compactifying the surface defect in the $6d$ theory associated to the $n$-index antisymmetric irrep $\Lambda^n$ of $\mathfrak{su}(2n)$. Notice that the square of this spinor gauge Wilson line is screened in the $5d$ theory, since the squared line has a trivial charge under the center of simply connected group $Spin(2n+1)$. This is consistent with its $6d$ lift as the square of the $\Lambda^n$ surface defect has trivial charge under the center of simply connected group $SU(2n)$, and hence the squared surface defect is screened in the $6d$ theory.

- Compactifying $6d$ $\mathfrak{g} = \mathfrak{so}(2n)$; $n \geq 5$ theory with $\mathbb{Z}_2$ outer-automorphism twist results in $5d$ $\mathfrak{h}_o^\vee = \mathfrak{sp}(n-1)$ gauge theory. The gauge Wilson lines of the $5d$ theory can be generated by taking products of the gauge Wilson line in the fundamental irrep of $\mathfrak{sp}(n-1)$ gauge algebra. This gauge Wilson line is produced by compactifying the surface defect in the $6d$ theory associated to the vector irrep of $\mathfrak{so}(2n)$. Notice that the square of this fundamental gauge Wilson line is screened in the $5d$ theory, since the squared line has a trivial charge under the center of simply connected group $Sp(n-1)$. This is consistent with its $6d$ lift as the square of the vector surface defect has trivial charge under the center of simply connected group $Spin(2n)$, and hence the squared surface defect is screened in the $6d$ theory.

- Compactifying $6d$ $\mathfrak{g} = \mathfrak{so}(8)$ theory with a $\mathbb{Z}_2$ outer-automorphism twist results in $5d$ $\mathfrak{h}_o^\vee = \mathfrak{sp}(n-1)$ gauge theory with discrete theta angle 0. There are three different $\mathbb{Z}_2$ twists[15], which can be characterized by the **8**-dimensional irrep $R$ of $\mathfrak{so}(8)$ left invariant by $o$. $R$ can be either the vector irrep, or the spinor irrep, or the cospinor irrep. The gauge Wilson lines of the $5d$ theory can be generated by taking products of the gauge Wilson line in the fundamental irrep of $\mathfrak{sp}(3)$ gauge algebra. This gauge Wilson line is produced by compactifying the surface defect in the $6d$ theory associated to the irrep

---

[15]Actually these three twists are all equivalent for $6d$ to $5d$ compactification as they are related to each other by gauge transformations for background discrete gauge field associated to the $\mathcal{O}_{\mathfrak{g}}$ bundle. When we consider to $6d$ to $4d$ compactifications, these three twists become apriori inequivalent (but can be equivalent depending on the global structure of the Riemann surface and other twists). For this reason, we consider all three of them here.

Table 1: Various data used throughout the paper. $\mathfrak{g}$ is a Lie algebra of $A, D, E$ type that specifies the $(2,0)$ theory. $\mathcal{O}_{\mathfrak{g}}$ denotes the outer-automorphism group of $\mathfrak{g}$. A dash denotes a trivial $\mathcal{O}_{\mathfrak{g}}$. $S_3$ denotes the permutation group of 3 objects. $\langle o \rangle$ denotes the subgroup of $\mathcal{O}_{\mathfrak{g}}$ generated by the outer-automorphism $o$. A dash denotes a trivial choice of $o$. $\mathbb{Z}_2^v, \mathbb{Z}_2^s, \mathbb{Z}_2^c$ are the subgroups of $\mathcal{O}_{\mathfrak{g}}$ generated by outer-automorphisms that leave invariant the vector, spinor and co-spinor irreps of $\mathfrak{so}(8)$ respectively. $\mathfrak{h}_o$ is the subalgebra of $\mathfrak{g}$ left invariant by $o$. $\mathfrak{h}_o^\vee$ is Langlands dual of $\mathfrak{h}_o$. $\mathcal{R}_o^\vee$ is a representation of $\mathfrak{h}_o^\vee$ and $\mathsf{R}_o$ is a representation of $\mathfrak{g}$ that descends to $\mathcal{R}_o^\vee$. $\mathsf{F}$ denotes the fundamental irrep of $\mathfrak{su}(n), \mathfrak{sp}(n)$, the vector irrep of $\mathfrak{so}(n)$, the 27-dimensional irrep of $\mathfrak{e}_6$, the 56-dimensional irrep of $\mathfrak{e}_7$, the 26-dimensional irrep of $\mathfrak{f}_4$, and the 7-dimensional irrep of $\mathfrak{g}_2$. $\Lambda^m$ denotes the $m$-index antisymmetric irrep of $\mathfrak{su}(n)$. $\mathsf{S}$ denotes a spinor irrep of $\mathfrak{so}(n)$ and $\mathsf{C}$ denotes the co-spinor irrep of $\mathfrak{so}(2n)$. $\mathsf{A}$ denotes the adjoint irrep. Note that for a trivial choice of $o$, we have $\mathfrak{g} = \mathfrak{h}_o = \mathfrak{h}_o^\vee$ and $\mathcal{R}_o^\vee = \mathsf{R}_o$. For $\mathfrak{g} = \mathfrak{so}(4n)$ and trivial $o$ case, we also define $\mathcal{R}_{o,s}^\vee = \mathsf{S}$ and $\mathcal{R}_{o,c}^\vee = \mathsf{C}$ by splitting $\mathcal{R}_o^\vee = \mathsf{S} \oplus \mathsf{C}$.

| $\mathfrak{g}$ | $\mathcal{O}_{\mathfrak{g}}$ | $\langle o \rangle$ | $\mathfrak{h}_o$ | $\mathfrak{h}_o^\vee$ | $\mathcal{R}_o^\vee$ | $\mathsf{R}_o$ |
|---|---|---|---|---|---|---|
| $\mathfrak{su}(2)$ | — | — | $\mathfrak{su}(2)$ | $\mathfrak{su}(2)$ | $\mathsf{F}$ | $\mathsf{F}$ |
| $\mathfrak{su}(2n);\ n \geq 2$ | $\mathbb{Z}_2$ | — | $\mathfrak{su}(2n)$ | $\mathfrak{su}(2n)$ | $\mathsf{F}$ | $\mathsf{F}$ |
| | | $\mathbb{Z}_2$ | $\mathfrak{sp}(n)$ | $\mathfrak{so}(2n+1)$ | $\mathsf{S}$ | $\Lambda^n$ |
| $\mathfrak{su}(2n+1)$ | $\mathbb{Z}_2$ | — | $\mathfrak{su}(2n+1)$ | $\mathfrak{su}(2n+1)$ | $\mathsf{F}$ | $\mathsf{F}$ |
| | | $\mathbb{Z}_2$ | $\mathfrak{so}(2n+1)$ | $\mathfrak{sp}(n)$ | $\mathsf{F}$ | $\mathsf{A}$ |
| $\mathfrak{so}(4n+2)$ | $\mathbb{Z}_2$ | — | $\mathfrak{so}(4n+2)$ | $\mathfrak{so}(4n+2)$ | $\mathsf{S}$ | $\mathsf{S}$ |
| | | $\mathbb{Z}_2$ | $\mathfrak{so}(4n+1)$ | $\mathfrak{sp}(2n)$ | $\mathsf{F}$ | $\mathsf{F}$ |
| $\mathfrak{so}(4n);\ n \geq 3$ | $\mathbb{Z}_2$ | — | $\mathfrak{so}(4n)$ | $\mathfrak{so}(4n)$ | $\mathsf{S} \oplus \mathsf{C}$ | $\mathsf{S} \oplus \mathsf{C}$ |
| | | $\mathbb{Z}_2$ | $\mathfrak{so}(4n-1)$ | $\mathfrak{sp}(2n-1)$ | $\mathsf{F}$ | $\mathsf{F}$ |
| $\mathfrak{so}(8)$ | $S_3$ | — | $\mathfrak{so}(8)$ | $\mathfrak{so}(8)$ | $\mathsf{S} \oplus \mathsf{C}$ | $\mathsf{S} \oplus \mathsf{C}$ |
| | | $\mathbb{Z}_2^v$ | $\mathfrak{so}(7)$ | $\mathfrak{sp}(3)$ | $\mathsf{F}$ | $\mathsf{F}$ |
| | | $\mathbb{Z}_2^s$ | $\mathfrak{so}(7)$ | $\mathfrak{sp}(3)$ | $\mathsf{F}$ | $\mathsf{S}$ |
| | | $\mathbb{Z}_2^c$ | $\mathfrak{so}(7)$ | $\mathfrak{sp}(3)$ | $\mathsf{F}$ | $\mathsf{C}$ |
| | | $\mathbb{Z}_3$ | $\mathfrak{g}_2$ | $\mathfrak{g}_2$ | $\mathsf{F}$ | $\mathsf{A}$ |
| $\mathfrak{e}_6$ | $\mathbb{Z}_2$ | — | $\mathfrak{e}_6$ | $\mathfrak{e}_6$ | $\mathsf{F}$ | $\mathsf{F}$ |
| | | $\mathbb{Z}_2$ | $\mathfrak{f}_4$ | $\mathfrak{f}_4$ | $\mathsf{F}$ | $\mathsf{A}$ |
| $\mathfrak{e}_7$ | — | — | $\mathfrak{e}_7$ | $\mathfrak{e}_7$ | $\mathsf{F}$ | $\mathsf{F}$ |
| $\mathfrak{e}_8$ | — | — | $\mathfrak{e}_8$ | $\mathfrak{e}_8$ | $\mathsf{A}$ | $\mathsf{A}$ |

$R$ of $\mathfrak{so}(8)$. Notice that the square of this fundamental gauge Wilson line is screened in the 5$d$ theory, since the squared line has a trivial charge under the center of simply connected group $Sp(3)$. This is consistent with its 6$d$ lift as the square of surface defect characterized by irrep $R$ has trivial charge under the center of simply connected group $Spin(8)$, and hence the squared surface defect is screened in the 6$d$ theory.

- Compactifying 6$d$ $\mathfrak{g} = \mathfrak{e}_6$ theory with $\mathbb{Z}_2$ outer-automorphism twist results in 5$d$ $\mathfrak{h}_o^\vee = \mathfrak{f}_4$ gauge theory. The gauge Wilson lines of the 5$d$ theory can be generated by taking products of the gauge Wilson line in the **26**-dimensional irrep of $\mathfrak{f}_4$ gauge algebra. This gauge Wilson line is produced by compactifying the surface defect in the 6$d$ theory associated to the adjoint irrep of $\mathfrak{e}_6$. Notice that this gauge Wilson line in the **26**-dimensional irrep is screened in the 5$d$ theory, since the center of simply connected group $F_4$ is trivial and hence this irrep has trivial charge under the center. This is consistent with its 6$d$ lift as

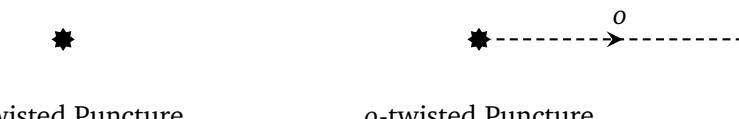

Untwisted Puncture          $o$-twisted Puncture

Figure 3: An untwisted puncture vs. a twisted puncture. An untwisted puncture is a genuine codimension-2 defect in the 6$d$ theory that lives at a point on the Riemann surface used to compactify the 6$d$ theory down to 4$d$. An $o$-twisted puncture is a non-genuine codimension-2 defect that is constrained to live at the end of a codimension-1 *topological* defect corresponding to an element $o$ of the $\mathcal{O}_\mathfrak{g}$ 0-form symmetry group of the 6$d$ theory. The codimension-2 defect lives at a point on the Riemann surface which lies at the end of an open line on the Riemann surface along which the codimension-1 topological operator is inserted. We refer to such a line where a codimension-1 topological operator (corresponding to an element of $\mathcal{O}_\mathfrak{g}$) is inserted as an outer-automorphism twist line.

the adjoint surface defect has trivial charge under the center of simply connected group $E_6$, and hence this surface defect is screened in the 6$d$ theory.

- Compactifying 6$d$ $\mathfrak{g} = \mathfrak{so}(8)$ theory with $\mathbb{Z}_3$ outer-automorphism twist results in 5$d$ $\mathfrak{h}_o^\vee = \mathfrak{g}_2$ gauge theory. The gauge Wilson lines of the 5$d$ theory can be generated by taking products of the gauge Wilson line in the **7**-dimensional irrep of $\mathfrak{g}_2$ gauge algebra. This gauge Wilson line is produced by compactifying the surface defect in the 6$d$ theory associated to the adjoint irrep of $\mathfrak{so}(8)$. Notice that this gauge Wilson line in the **7**-dimensional irrep is screened in the 5$d$ theory, since the center of simply connected group $G_2$ is trivial and hence this irrep has trivial charge under the center. This is consistent with its 6$d$ lift as the adjoint surface defect has trivial charge under the center of simply connected group $Spin(8)$, and hence this surface defect is screened in the 6$d$ theory.

- Compactifying 6$d$ $\mathfrak{g} = \mathfrak{su}(2n+1)$ theory with $\mathbb{Z}_2$ outer-automorphism twist results in 5$d$ $\mathfrak{h}_o^\vee = \mathfrak{sp}(n)$ gauge theory with discrete theta angle $\pi$ [6]. The gauge Wilson lines of the 5$d$ theory can be generated by taking products of the gauge Wilson line in the fundamental irrep of $\mathfrak{sp}(n)$ gauge algebra. This gauge Wilson line is produced by compactifying the surface defect in the 6$d$ theory associated to the adjoint irrep of $\mathfrak{su}(2n+1)$. Notice that the adjoint surface defect is screened in the 6$d$ theory, and so consistency requires that the fundamental gauge Wilson line must also be screened in the 5$d$ theory even though it carries a non-trivial charge under the $\mathbb{Z}_2$ center of the simply connected group $Sp(n)$. In fact, it is known that the 5$d$ $\mathcal{N} = 2$ $\mathfrak{sp}(n)$ gauge theory with discrete theta angle $\pi$ has a BPS instanton particle carrying a non-trivial charge under the center of $Sp(n)$ (see for example [7]). The local operator associated to this particle is a non-genuine local operator that can be inserted at the end of a gauge Wilson line in fundamental irrep of $\mathfrak{sp}(n)$. Thus, the fundamental gauge Wilson line of the 5$d$ theory is also screened, and the 5$d$ and 6$d$ pictures are consistent with each other.

Let us denote as $\mathcal{R}_o^\vee$ the representation of $\mathfrak{h}_o^\vee$ generating all the other representations via tensor products. The representation of $\mathfrak{g}$ corresponding to $\mathcal{R}_o^\vee$ is denoted by $\mathsf{R}_o$. See Table 1.

## 3.2 Local Operators in 4$d$ Arising from Punctures

Consider a regular puncture $\mathcal{P}$ for $(2,0)$ theory of type $\mathfrak{g}$ living at the end of an $o$ outer-automorphism twist line. An untwisted regular puncture corresponds to the case when $o$ is

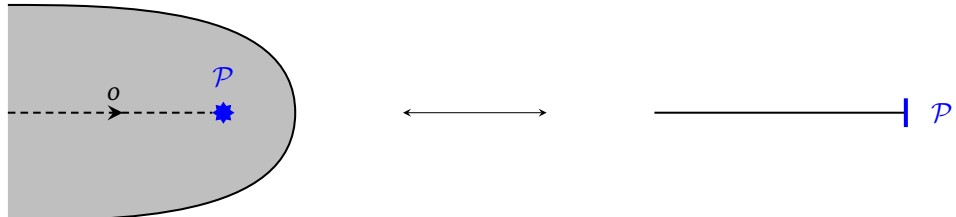

$6d$ $\mathcal{N} = (2,0)$ theory of type $\mathfrak{g}$ with $o$-twisted puncture $\mathcal{P}$

$5d$ $\mathcal{N} = 2$ SYM with gauge algebra $\mathfrak{h}_o^\vee$ and boundary condition $\mathcal{P}$

Figure 4: Compactifying $6d$ $\mathcal{N} = (2,0)$ theory of type $\mathfrak{g}$ on a cigar-like non-compact surface with an $o$-twisted puncture $\mathcal{P}$ placed at the tip of the cigar is equivalent to $5d$ $\mathcal{N} = 2$ SYM theory on a half-line with gauge algebra $\mathfrak{h}_o^\vee$ and a boundary condition associated to $\mathcal{P}$ (which, by an abuse of notation, we label by $\mathcal{P}$ in the figure) placed at the end of the half-line.

trivial. See Figure 3. The puncture $\mathcal{P}$ is associated to a homomorphism

$$\rho : \mathfrak{su}(2) \to \mathfrak{h}_o^\vee , \tag{10}$$

where $\mathfrak{h}_o^\vee = \mathfrak{g}$ if $o$ is trivial. We can regard the puncture as a boundary condition in the $5d$ $\mathcal{N} = 2$ $\mathfrak{h}_o^\vee$ SYM. See Figure 4. This boundary condition is such that the gauge algebra $\mathfrak{h}_o^\vee$ reduces to a flavor algebra $\mathfrak{f}_\mathcal{P}$ at the $4d$ boundary where $\mathfrak{f}_\mathcal{P}$ is the commutant in $\mathfrak{h}_o^\vee$ of the image $\rho(\mathfrak{su}(2)) \subseteq \mathfrak{h}_o^\vee$. Let us write $\mathfrak{f}_\mathcal{P} = \mathfrak{f}_{na,\mathcal{P}} \oplus \mathfrak{u}(1)_\mathcal{P}^a$ where $\mathfrak{f}_{na,\mathcal{P}}$ is a non-abelian semi-simple Lie algebra and $\mathfrak{u}(1)_\mathcal{P}^a$ is the abelian part. We associate a group $F_\mathcal{P} = F_{na,\mathcal{P}} \times U(1)_\mathcal{P}^a$ to such a puncture $\mathcal{P}$. $F_\mathcal{P}$ is a group whose associated Lie algebra is $\mathfrak{f}_\mathcal{P}$, where $F_{na,\mathcal{P}}$ is the simply connected group associated to $\mathfrak{f}_{na,\mathcal{P}}$ and $U(1)_\mathcal{P}^a$ is a group whose associated Lie algebra is $\mathfrak{u}(1)_\mathcal{P}^a$. We denote the center of $F_\mathcal{P}$ as $Z_{F,\mathcal{P}}$ and its Pontryagin dual as $\widehat{Z}_{F,\mathcal{P}}$.

A representation $R_o^\vee$ of $\mathfrak{h}_o^\vee$ becomes a representation $R_{o,\mathcal{P}}^\vee$ of $\mathfrak{f}_\mathcal{P}$, where $R_{o,\mathcal{P}}^\vee$ is simply the representation $R_o^\vee$ viewed from the point of view of $\mathfrak{f}_\mathcal{P} \subseteq \mathfrak{h}_o^\vee$.

If $\mathfrak{h}_o^\vee \neq \mathfrak{so}(4n)$, the $5d$ theory contains a local operator in the representation

$$\mathcal{R}_o^\vee \otimes \overline{\mathcal{R}}_o^\vee \tag{11}$$

of $\mathfrak{h}_o^\vee$ where $\overline{\mathcal{R}}_o^\vee$ is the complex conjugate of $\mathcal{R}_o^\vee$. This is because the representation $\mathcal{R}_o^\vee \otimes \overline{\mathcal{R}}_o^\vee$ is uncharged under the center of the simply connected group $H_o^\vee$ associated to the algebra $\mathfrak{h}_o^\vee$, and hence the representation $\mathcal{R}_o^\vee \otimes \overline{\mathcal{R}}_o^\vee$ can be generated from tensor products of adjoint representation. For $\mathfrak{h}_o^\vee = \mathfrak{so}(4n)$, we have $\mathcal{R}_o^\vee = \mathsf{S} \oplus \mathsf{C}$, i.e. $\mathcal{R}_o^\vee$ is the direct sum of spinor and cospinor irreps of $\mathfrak{so}(4n)$. Let us define $\mathcal{R}_{o,s}^\vee = \mathsf{S}$ and $\mathcal{R}_{o,c}^\vee = \mathsf{C}$. For $\mathfrak{h}_o^\vee = \mathfrak{so}(4n)$, the $5d$ theory contains a local operator transforming in the representation

$$\left( \mathcal{R}_{o,s}^\vee \otimes \overline{\mathcal{R}}_{o,s}^\vee \right) \oplus \left( \mathcal{R}_{o,c}^\vee \otimes \overline{\mathcal{R}}_{o,c}^\vee \right) \tag{12}$$

of $\mathfrak{h}_o^\vee = \mathfrak{so}(4n)$.

For $\mathfrak{h}_o^\vee \neq \mathfrak{so}(4n)$, when this local operator is moved to the $4d$ boundary associated to $\mathcal{P}$, then it becomes a local operator $\mathcal{O}_\mathcal{P}$ charged in representation

$$\mathcal{R}_{o,\mathcal{P}}^\vee \otimes \overline{\mathcal{R}}_{o,\mathcal{P}}^\vee \tag{13}$$

of the flavor algebra $\mathfrak{f}_\mathcal{P}$. On the other hand, for $\mathfrak{h}_o^\vee = \mathfrak{so}(4n)$, $\mathcal{O}_\mathcal{P}$ is charged in representation

$$\left( \mathcal{R}_{o,s,\mathcal{P}}^\vee \otimes \overline{\mathcal{R}}_{o,s,\mathcal{P}}^\vee \right) \oplus \left( \mathcal{R}_{o,c,\mathcal{P}}^\vee \otimes \overline{\mathcal{R}}_{o,c,\mathcal{P}}^\vee \right) \tag{14}$$

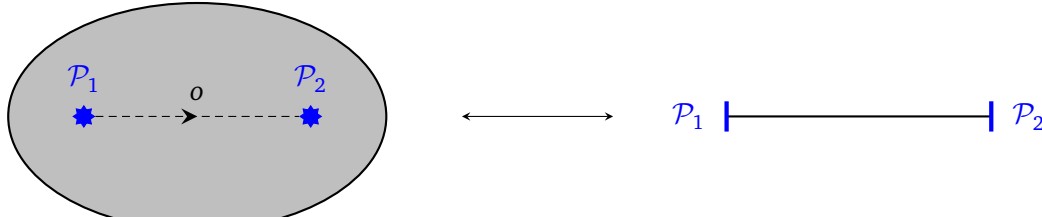

Figure 5: Compactifying 6$d$ $\mathcal{N} = (2,0)$ theory of type $\mathfrak{g}$ on a sphere-like surface with an $o$-twisted puncture $\mathcal{P}_1$ and $o^{-1}$-twisted puncture is equivalent to compactifying 5$d$ $\mathcal{N} = 2$ SYM theory with gauge algebra $\mathfrak{h}_o^\vee$ on an interval with boundary conditions associated to $\mathcal{P}_1$ and $\mathcal{P}_2$ inserted at the two ends of the interval.

of the flavor algebra $\mathfrak{f}_\mathcal{P}$.

The representations (13) and (14) apriori contain a number of irreps of $\mathfrak{f}_\mathcal{P}$. The local operator $\mathcal{O}_\mathcal{P}$ decomposes into local operators transforming in these irreps. The charges under $Z_{F,\mathcal{P}}$ of these local operators generate a sub-lattice $Y_{F,\mathcal{P}}$ of $\widehat{Z}_{F,\mathcal{P}}$.

Consider a 4$d$ $\mathcal{N} = 2$ class S theory arising from the compactification of a 6$d$ $\mathcal{N} = (2,0)$ theory carrying regular (untwisted and twisted) punctures $\mathcal{P}_i$ and closed twist lines. Then, the manifest flavor symmetry of the 4$d$ theory is[16] $\mathfrak{f} = \bigoplus_i \mathfrak{f}_{\mathcal{P}_i}$ which has an associated lattice of charges $\widehat{Z}_F = \bigoplus_i \widehat{Z}_{F,\mathcal{P}_i}$. From the above consideration, we find that the sub-lattice

$$Y_F' = \bigoplus_i Y_{F,\mathcal{P}_i} \subseteq \widehat{Z}_F \tag{15}$$

of charges is realized by local operators in the 4$d$ theory.

### 3.3 Local Operators in 4$d$ Arising from Surface Defects in 6$d$

Notice that the local operators discussed above carry charges only under the center associated to a single puncture. Now we turn to a discussion of other local operators that can carry charges under centers associated to multiple punctures. We start with a simple situation in which we compactify 6$d$ type $\mathfrak{g}$ $(2,0)$ theory on a sphere with two punctures $\mathcal{P}_1$ and $\mathcal{P}_2$, such that $\mathcal{P}_1$ lives at the end of an $o$ twist line while $\mathcal{P}_2$ lives at the end of an $o^{-1}$ twist line. Then we have an $o$ twist line running from $\mathcal{P}_1$ to $\mathcal{P}_2$. See Figure 5. This setup can be equivalently represented as a compactification of 5$d$ $\mathcal{N} = 2$ SYM with gauge algebra $\mathfrak{h}_o^\vee$ on a segment with the boundary condition corresponding to $\mathcal{P}_1$ placed at one boundary of the segment, and the boundary condition corresponding to $\mathcal{P}_2$ placed at the other boundary of the segment. The gauge Wilson line in representation $\mathcal{R}_o^\vee$ of $\mathfrak{h}_\vee^o$ can be inserted along the segment such that its end points lie at the two boundaries of the segment. This Wilson line gives rise to a local operator $\mathcal{O}_{\mathcal{P}_1,\mathcal{P}_2}$ in the 4$d$ theory charged in representation

$$\mathcal{R}_{o,\mathcal{P}_1}^\vee \otimes \mathcal{R}_{o,\mathcal{P}_2}^\vee \tag{16}$$

of the flavor algebra $\mathfrak{f}_{\mathcal{P}_1} \oplus \mathfrak{f}_{\mathcal{P}_2}$. In a similar way, the local operators associated to (13) and (14) can also be understood as the contribution of $\mathcal{R}_o^\vee$ Wilson line whose both ends lie at the same boundary $\mathcal{P}$. Note that, unlike (13), there is no complex conjugate appearing in the above equation (16). This is due to the fact that a Wilson line traveling from the boundary $\mathcal{P}$ to itself is akin to a Wilson line traveling from $\mathcal{P}_1 = \mathcal{P}$ to $\mathcal{P}_2 = \overline{\mathcal{P}}$, where $\overline{\mathcal{P}}$ is the boundary obtained by changing the orientation of $\mathcal{P}$ (so that it lies at the other end of the segment).

---

[16]Note the use of $\mathfrak{f}$ to denote the manifest flavor symmetry. This usage is in contrast with the earlier subsections, where $\mathfrak{f}$ is used to denote the full flavor symmetry algebra and manifest flavor symmetry algebra is denoted as $\mathfrak{f}_m$ instead. We hope that in the rest of the paper, it would be clear from the context whether $\mathfrak{f}$ refers to the manifest or to the full flavor symmetry algebra.

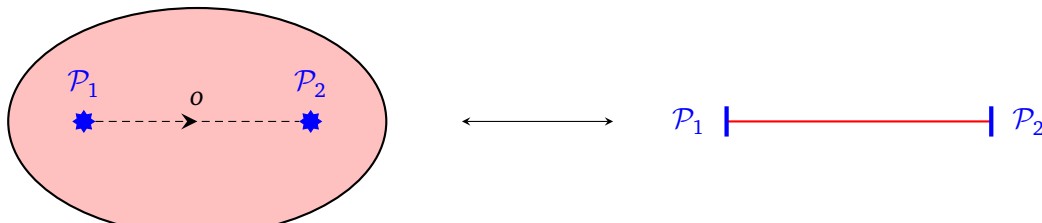

Figure 6: Inserting a Wilson line in rep $\mathcal{R}_o^\vee$ stretched between the two boundaries (denoted by red) in the 5$d$ theory is equivalent to compactifying the R$_o$ surface defect along the whole sphere (denoted by red background) in the 6$d$ theory.

The above $\mathcal{R}_o^\vee$ Wilson line inserted between the two boundaries corresponding to $\mathcal{P}_1$ and $\mathcal{P}_2$ lifts in the 6$d$ theory to a R$_o$ surface defect wrapped along the whole sphere. See Figure 6. It is possible to wrap this surface defect because the representation R$_o$ of $\mathfrak{g}$ is left invariant by the action of $o$. We can now generalize to a more general compactification as follows. Along the whole Riemann surface, we can wrap a surface defect of the 6$d$ theory that is left invariant by all outer-automorphism twist lines. From this we obtain a local operator $\mathcal{O}$ in the 4$d$ theory transforming in a representation $\mathcal{R}$ of the flavor symmetry algebra $\mathfrak{f} = \bigoplus_i \mathfrak{f}_{\mathcal{P}_i}$. The precise form of $\mathcal{R}$ depends on the type of compactification and is discussed later in this subsection. We can decompose this representation as

$$\mathcal{R} = \bigoplus_\alpha \mathcal{R}_\alpha \,, \tag{17}$$

such that each $\mathcal{R}_\alpha$ is an irrep of $\mathfrak{f}$. Then the local operator $\mathcal{O}$ decomposes into local operators $\mathcal{O}_\alpha$ transforming in irreps $\mathcal{R}_\alpha$ which generate a lattice of charges

$$\widetilde{Y}_F \subseteq \widehat{Z}_F = \bigoplus_i \widehat{Z}_{F,\mathcal{P}_i} \,. \tag{18}$$

Thus, the total sub-lattice $Y_F$ of $\widehat{Z}_F$ formed by charges of local operators appearing in the 4$d$ theory is

$$Y_F = \left\langle Y'_F \cup \widetilde{Y}_F \right\rangle \,, \tag{19}$$

which is the sublattice generated by the union of sublattices $Y'_F$ and $\widetilde{Y}_F$. See (15) for the definition of the sublattice $Y'_F$. Now one can use this $Y_F$ to compute the flavor symmetry group of the 4$d$ theory as in (4).

We now turn to a discussion of the precise form of the representation $\mathcal{R}$ for various kinds of compactifications:

**Untwisted :** If all the regular punctures $\mathcal{P}_i$ are untwisted and there are no (homologically non-trivial) closed twist lines, then we can wrap the surface defect carrying representation $\mathcal{R}_o^\vee$ corresponding to $\mathfrak{h}_o^\vee = \mathfrak{g}$ along the whole Riemann surface. For $\mathfrak{g} \neq \mathfrak{so}(4n)$, this leads to a local operator $\mathcal{O}$ transforming in representation

$$\mathcal{R} = \bigotimes_i \mathcal{R}_{o,\mathcal{P}_i}^\vee \tag{20}$$

of the flavor symmetry algebra $\mathfrak{f} = \bigoplus_i \mathfrak{f}_{\mathcal{P}_i}$. For $\mathfrak{g} = \mathfrak{so}(4n)$, this leads to a local operator $\mathcal{O}$ transforming in representation

$$\mathcal{R} = \left( \bigotimes_i \mathcal{R}_{o,s,\mathcal{P}_i}^\vee \right) \oplus \left( \bigotimes_i \mathcal{R}_{o,c,\mathcal{P}_i}^\vee \right) \tag{21}$$

of the flavor symmetry algebra $\mathfrak{f} = \bigoplus_i \mathfrak{f}_{\mathcal{P}_i}$.

**Single type of twist lines :** Consider the situation when the outer-automorphism elements associated to all the closed twist lines and the open twist lines associated to twisted regular punctures lie in a $\mathbb{Z}_2$ or a $\mathbb{Z}_3$ subgroup of $\mathcal{O}_\mathfrak{g}$. Then all the twist lines are associated to the same $\mathfrak{h}_o^\vee$. Let us define two index sets $\mathcal{T}$ and $\mathcal{U}$ by the following criteria. A puncture $\mathcal{P}_i$ such that $i \in \mathcal{T}$ is a twisted regular puncture, while a puncture $\mathcal{P}_i$ such that $i \in \mathcal{U}$ is an untwisted regular puncture. In such a situation, we can wrap the surface defect carrying representation $\mathsf{R}_o$ of $\mathfrak{g}$ corresponding to the representation $\mathcal{R}_o^\vee$ of $\mathfrak{h}_o^\vee$ along the whole Riemann surface. This leads to a local operator $\mathcal{O}$ transforming in representation

$$\mathcal{R} = \bigotimes_{i \in \mathcal{T}} \mathcal{R}_{o,\mathcal{P}_i}^\vee \bigotimes_{i \in \mathcal{U}} \mathsf{R}_{o,\mathcal{P}_i} \tag{22}$$

of the flavor symmetry algebra $\mathfrak{f} = \bigoplus_i \mathfrak{f}_{\mathcal{P}_i} = \bigoplus_{i \in \mathcal{T}} \mathfrak{f}_{\mathcal{P}_i} \bigoplus_{i \in \mathcal{U}} \mathfrak{f}_{\mathcal{P}_i}$. Here $\mathsf{R}_{o,\mathcal{P}_i}$ is the representation $\mathsf{R}_o$ of $\mathfrak{g}$ viewed from the point of view of the flavor symmetry algebra $\mathfrak{f}_{\mathcal{P}_i} \subseteq \mathfrak{g}$ associated to an untwisted regular puncture $\mathcal{P}_i$.

**Multiple types of twist lines :** Consider the situation when the outer-automorphism elements associated to all the closed twist lines and the open twist lines associated to twisted regular punctures lie in either multiple $\mathbb{Z}_2$ subgroups of $\mathcal{O}_\mathfrak{g}$, or a $\mathbb{Z}_2$ and $\mathbb{Z}_3$ subgroup of $\mathcal{O}_\mathfrak{g}$, or a $\mathbb{Z}_3$ subgroup and multiple $\mathbb{Z}_2$ subgroups of $\mathcal{O}_\mathfrak{g}$. This kind of a situation is possible only for $\mathfrak{g} = \mathfrak{so}(8)$ for which we have $\mathcal{O}_\mathfrak{g} = S_3$ the permutation group of three objects. Let us define index sets $\mathcal{U}$, $\mathcal{T}_3$ and $\mathcal{T}_2$ by the following criteria. A puncture $\mathcal{P}_i$ such that $i \in \mathcal{U}$ is an untwisted regular puncture, a puncture $\mathcal{P}_i$ such that $i \in \mathcal{T}_3$ is a twisted regular puncture living at the end of a twist line carrying an outer automorphism element $o$ of order three, and a puncture $\mathcal{P}_i$ such that such that $i \in \mathcal{T}_2$ is a twisted regular puncture living at the end of a twist line carrying an outer automorphism element $o$ of order two. In such a situation, we can wrap the surface defect carrying the adjoint representation of $\mathfrak{g} = \mathfrak{so}(8)$ along the whole Riemann surface. Let us label the adjoint representation by $\mathsf{R}$. This leads to a local operator $\mathcal{O}$ transforming in representation

$$\mathcal{R} = \bigotimes_{i \in \mathcal{T}_3} \mathcal{R}_{o,\mathcal{P}_i}^\vee \bigotimes_{i \in \mathcal{U}} \mathsf{R}_{\mathcal{P}_i} \bigotimes_{i \in \mathcal{T}_2} \mathsf{R}_{o,\mathcal{P}_i}^\vee \tag{23}$$

of the flavor symmetry algebra $\mathfrak{f} = \bigoplus_i \mathfrak{f}_{\mathcal{P}_i} = \bigoplus_{i \in \mathcal{T}_3} \mathfrak{f}_{\mathcal{P}_i} \bigoplus_{i \in \mathcal{U}} \mathfrak{f}_{\mathcal{P}_i} \bigoplus_{i \in \mathcal{T}_2} \mathfrak{f}_{\mathcal{P}_i}$. Here $\mathsf{R}_{\mathcal{P}_i}$ is the adjoint representation $\mathsf{R}$ of $\mathfrak{g} = \mathfrak{so}(8)$ viewed from the point of view of the flavor symmetry algebra $\mathfrak{f}_{\mathcal{P}_i} \subseteq \mathfrak{so}(8)$ associated to an untwisted regular puncture $\mathcal{P}_i$. The representation $\mathsf{R}_{o,\mathcal{P}_i}^\vee$ is the 2-index antisymmetric irrep of $\mathfrak{h}_o^\vee = \mathfrak{sp}(3)$ (associated to a $\mathbb{Z}_2$ twisted puncture) viewed from the point of view of the flavor symmetry algebra $\mathfrak{f}_{\mathcal{P}_i} \subseteq \mathfrak{sp}(3)$ associated to a $\mathbb{Z}_2$ twisted regular puncture $\mathcal{P}_i$.

# 4 Illustrative Examples and Consistency Checks

In this section, we illustrate using various examples the above discussed procedure for determining manifest flavor symmetry groups of $4d$ $\mathcal{N} = 2$ Class S theories. See the end of Section 1 for an overview of this section.

We label punctures using the notation of [8–18], [4] which captures the homomorphism (10) associated to each puncture. We refer the reader to these papers for more details about the notation and corresponding homomorphisms.

## 4.1 Free-field Fixtures

**Simplest example – $T_2$ theory :**

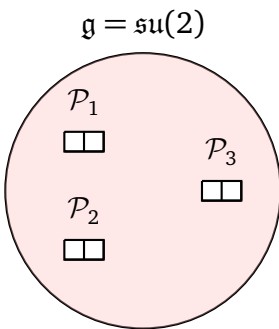

Consider compactifying $\mathfrak{g} = \mathfrak{su}(2)$ 6d $\mathcal{N} = (2,0)$ theory on a sphere with three regular punctures. Let us label the three punctures by $\mathcal{P}_i$ with $i = 1, 2, 3$. The puncture $\mathcal{P}_i$ has an $\mathfrak{f}_{\mathcal{P}_i} = \mathfrak{su}(2)_i$ flavor symmetry algebra associated to it. The representation $\mathcal{R}_o^\vee$ is the fundamental representation of $\mathfrak{h}_o^\vee = \mathfrak{g} = \mathfrak{su}(2)$ and thus $\mathcal{R}_{o,\mathcal{P}_i}^\vee$ is the fundamental representation $\mathsf{F}_i$ of $\mathfrak{su}(2)_i$. The local operators in the 4d theory contributed locally by $\mathcal{P}_i$ are generated by the representation (13) which is $\mathsf{F}_i \otimes \mathsf{F}_i$ which has a trivial charge under the center $Z_{F,\mathcal{P}_i} \simeq \mathbb{Z}_2$ of the simply connected group $F_{\mathcal{P}_i} = SU(2)_i$ associated to $\mathfrak{su}(2)_i$. Thus $Y_{F,\mathcal{P}_i} = 0$ and hence $Y_F' = 0$. We still need to consider 4d local operators arising from the compactification of surface defect in fundamental representation of $\mathfrak{g} = \mathfrak{su}(2)$ along the sphere. This leads to a local operator in 4d transforming in representation (20)

$$\mathcal{R} = \mathsf{F}_1 \otimes \mathsf{F}_2 \otimes \mathsf{F}_3 \tag{24}$$

of $\mathfrak{su}(2)_1 \oplus \mathfrak{su}(2)_2 \oplus \mathfrak{su}(2)_3$. Thus $\widetilde{Y}_F \simeq \mathbb{Z}_2$ generated by the element $(1, 1, 1) \in \widehat{Z}_{F,\mathcal{P}_1} \oplus \widehat{Z}_{F,\mathcal{P}_2} \oplus \widehat{Z}_{F,\mathcal{P}_3}$. Since $Y_F' = 0$, we have $Y_F = \widetilde{Y}_F$, from which we compute that the *manifest* flavor symmetry group is

$$\mathcal{F} = \frac{SU(2)_1 \times SU(2)_2 \times SU(2)_3}{\mathbb{Z}_2^{1,2} \times \mathbb{Z}_2^{2,3}}, \tag{25}$$

where $\mathbb{Z}_2^{i,j}$ is the diagonal $\mathbb{Z}_2$ subgroup of $Z_{F,\mathcal{P}_i} \times Z_{F,\mathcal{P}_j}$.

We can confirm this result using the analysis of Section 2.2 since the resulting 4d $\mathcal{N} = 2$ theory admits a Lagrangian description. The 4d theory is a bunch of free hypers transforming as a half-hyper in trifundamental representation (24). Thus we have $\mathcal{M} = \widetilde{Y}_F$ which leads to precisely the same result (25).

**Bifundamental hyper :**

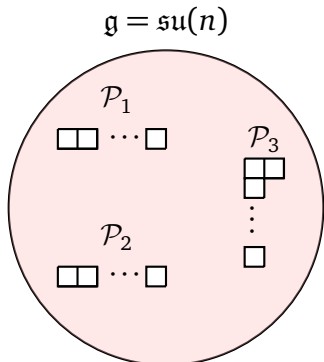

The puncture $\mathcal{P}_i$ has an $\mathfrak{f}_{\mathcal{P}_i} = \mathfrak{su}(n)_i$ flavor symmetry algebra associated to it for $i = 1, 2$, and $\mathcal{P}_3$ has $\mathfrak{f}_{\mathcal{P}_3} = \mathfrak{u}(1)$. The representation $\mathcal{R}_o^\vee$ is the fundamental representation of $\mathfrak{h}_o^\vee = \mathfrak{g} = \mathfrak{su}(n)$ and thus $\mathcal{R}_{o,\mathcal{P}_i}^\vee$ is the fundamental representation $\mathsf{F}_i$ of $\mathfrak{su}(n)_i$ for $i = 1, 2$. On the other hand, for the minimal puncture the representation $\mathcal{R}_{o,\mathcal{P}_3}^\vee = (n-1) \cdot \mathbf{1}_1 \oplus \mathbf{1}_{1-n}$ where $\mathbf{1}_q$ denotes the charge $q$ representation of $F_{\mathcal{P}_3} = U(1)$. From this we compute that, for $i = 1, 2$, we have $Y_{F,\mathcal{P}_i} = 0$ as $\mathcal{R}_{o,\mathcal{P}_i}^\vee \otimes \overline{\mathcal{R}}_{o,\mathcal{P}_i}^\vee = \mathsf{F}_i \otimes \overline{\mathsf{F}}_i$ has zero charge under $Z_{F,\mathcal{P}_i}$. On the other hand, because $\mathcal{R}_{o,\mathcal{P}_3}^\vee \otimes \overline{\mathcal{R}}_{o,\mathcal{P}_3}$ contains $F_{\mathcal{P}_3} = U(1)$ charges equal to $n, -n, 0$, we have $Y_{F,\mathcal{P}_3} = n\mathbb{Z}$ if we represent $\widehat{Z}_{F,\mathcal{P}_3} = \mathbb{Z}$. Thus, $Y_F'$ is generated by the element $(0, 0, n) \in \widehat{Z}_{F,\mathcal{P}_1} \oplus \widehat{Z}_{F,\mathcal{P}_2} \oplus \widehat{Z}_{F,\mathcal{P}_3}$.

The representation (20) becomes

$$\mathsf{F}_1 \otimes \mathsf{F}_2 \otimes \left( (n-1) \cdot \mathbf{1}_1 \oplus \mathbf{1}_{1-n} \right), \tag{26}$$

implying that $\widetilde{Y}_F$ is generated by the element $(1, 1, 1) \in \widehat{Z}_{F,\mathcal{P}_1} \oplus \widehat{Z}_{F,\mathcal{P}_2} \oplus \widehat{Z}_{F,\mathcal{P}_3}$. The other element $(1, 1, 1-n) \in \widehat{Z}_{F,\mathcal{P}_1} \oplus \widehat{Z}_{F,\mathcal{P}_2} \oplus \widehat{Z}_{F,\mathcal{P}_3}$ is included in $\widetilde{Y}_F$ because it is $(1-n)$ times the element $(1, 1, 1)$. Notice that $n$ times $(1, 1, 1)$ is the generator of $Y_F'$, so $Y_F'$ is a sub-lattice of $\widetilde{Y}_F$ and from (19) we have $Y_F = \widetilde{Y}_F$.

Now let us compute the manifest flavor symmetry group $\mathcal{F}$. We have

$$F = SU(n)_1 \times SU(n)_2 \times U(1), \tag{27}$$

with center $(\mathbb{Z}_n)_1 \times (\mathbb{Z}_n)_2 \times U(1)$. Let us represent the elements of $(\mathbb{Z}_n)_i$ as $\frac{k}{n}$ (mod 1) and the elements of $U(1)$ as living in $\mathbb{R}/\mathbb{Z}$. Then we are looking for elements $(\alpha, \beta, \gamma) \in (\mathbb{Z}_n)_1 \times (\mathbb{Z}_n)_2 \times U(1)$ such that its scalar product with $(1,1,1)$ is zero. That is, we are looking for solutions to

$$(1, 1, 1) \cdot (\alpha, \beta, \gamma) = 0. \tag{28}$$

The solutions to this equation form a group $\mathbb{Z}_n^{1,2} \times \mathbb{Z}_n^{2,3}$, where $\mathbb{Z}_n^{1,2}$ is the $\mathbb{Z}_n$ subgroup generated by the element $\left( \frac{1}{n}, -\frac{1}{n}, 0 \right)$ of the $(\mathbb{Z}_n)_1 \times (\mathbb{Z}_n)_2 \times U(1)$ center and $\mathbb{Z}_n^{2,3}$ factor is the $\mathbb{Z}_n$ subgroup generated by the element $\left( 0, \frac{1}{n}, -\frac{1}{n} \right)$ of the $\mathbb{Z}_n^1 \times \mathbb{Z}_n^2 \times U(1)$ center. Thus,

$$\mathcal{F} = \frac{SU(n)_1 \times SU(n)_2 \times U(1)}{\mathbb{Z}_n^{1,2} \times \mathbb{Z}_n^{2,3}} \tag{29}$$

is the manifest flavor group.

We can confirm this result using the analysis of Section 2.2 since the resulting $4d$ $\mathcal{N} = 2$ theory admits a Lagrangian description. The $4d$ theory is a bunch of free hypers transforming as a hyper in the bifundamental representation $\mathsf{F}_1 \otimes \mathsf{F}_2$ of the manifest flavor symmetry algebra $\mathfrak{su}(n)_1 \oplus \mathfrak{su}(n)_2$. The third manifest flavor algebra $\mathfrak{u}(1)$ rotates the bifundamental hyper. We can represent the $4d$ $\mathcal{N} = 2$ Lagrangian theory as a quiver diagram of the form

$$\begin{array}{c} \left[ \mathfrak{u}(1) \right] \\ \Big| \mathsf{F} \\ \left[ \mathfrak{su}(n)_1 \right] \overset{\mathsf{F}}{\rule{3em}{0.4pt}}\!\!\! \overset{\mathsf{F}}{\rule{3em}{0.4pt}} \left[ \mathfrak{su}(n)_2 \right] \end{array}, \tag{30}$$

where an algebra in brackets denotes a flavor algebra and $\mathsf{F}$ for $\mathfrak{u}(1)$ denotes its charge $+1$ representation. For this Lagrangian description, we have $\mathcal{M} = \widetilde{Y}_F$ which leads to precisely the same result (29).

**Example including non-minimal and non-maximal punctures :**

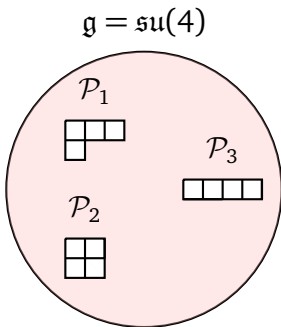

The puncture $\mathcal{P}_3$ has an $\mathfrak{f}_{\mathcal{P}_3} = \mathfrak{su}(4)$ flavor symmetry, $\mathcal{P}_2$ has $\mathfrak{f}_{\mathcal{P}_2} = \mathfrak{su}(2)_2$, and $\mathcal{P}_1$ has $\mathfrak{f}_{\mathcal{P}_1} = \mathfrak{su}(2)_1 \oplus \mathfrak{u}(1)$. The representation $\mathcal{R}_o^\vee$ is the fundamental representation of $\mathfrak{h}_o^\vee = \mathfrak{g} = \mathfrak{su}(4)$. We have $\mathcal{R}_{o,\mathcal{P}_3}^\vee = \mathsf{F}$ the fundamental representation of $\mathfrak{su}(4)$, $\mathcal{R}_{o,\mathcal{P}_2}^\vee = 2 \cdot \mathsf{F}$ with $\mathsf{F}$ being the fundamental representation of $\mathfrak{su}(2)_2$, and $\mathcal{R}_{o,\mathcal{P}_1}^\vee = \mathsf{F}_{-1} \oplus 2 \cdot \mathbf{1}_1$ where $\mathsf{F}_{-1}$ is the representation $\mathsf{F} \otimes \mathbf{1})-1$ of $\mathfrak{f}_{\mathcal{P}_1} = \mathfrak{su}(2)_1 \oplus \mathfrak{u}(1)$ and $\mathbf{1}_1$ is the representation $\mathbf{1} \otimes \mathbf{1})1$ of $\mathfrak{f}_{\mathcal{P}_1} = \mathfrak{su}(2)_1 \oplus \mathfrak{u}(1)$. From this we compute that $Y_{F,\mathcal{P}_i} = 0$ for $i = 2,3$ but $Y_{F,\mathcal{P}_1}$ is the sub-lattice generated by $(1,-2)$ in $\widehat{Z}_{F,\mathcal{P}_1} = (\mathbb{Z}/2\mathbb{Z}) \oplus \mathbb{Z}$. Thus, $Y_F'$ is generated by the element $(1,-2,0,0) \in \widehat{Z}_{F,\mathcal{P}_1} \oplus \widehat{Z}_{F,\mathcal{P}_2} \oplus \widehat{Z}_{F,\mathcal{P}_3}$.

The local operators in representation (20) imply that $\widetilde{Y}_F$ is generated by the elements $(0,1,1,1)$ and $(1,-1,1,1)$ in $\widehat{Z}_{F,\mathcal{P}_1} \oplus \widehat{Z}_{F,\mathcal{P}_2} \oplus \widehat{Z}_{F,\mathcal{P}_3}$. Notice that $Y_F' \subset \widetilde{Y}_F$ and so $Y_F = \widetilde{Y}_F$. Moreover, notice that the generators of $Y_F$ can also be chosen to be $(1,0,0,2)$ and $(0,1,1,1)$. Using $Y_F$, we compute that the manifest flavor symmetry group is

$$\mathcal{F} = \frac{SU(2)_1 \times U(1) \times SU(2)_2 \times SU(4)}{\mathbb{Z}_4 \times \mathbb{Z}_2}, \tag{31}$$

where $\mathbb{Z}_4$ factor is generated by the element $\left(\frac{1}{2}, -\frac{1}{4}, 0, \frac{1}{4}\right) \in Z_{F,\mathcal{P}_1} \oplus Z_{F,\mathcal{P}_2} \oplus Z_{F,\mathcal{P}_3} \simeq \mathbb{Z}_2 \times \mathbb{R}/\mathbb{Z} \times \mathbb{Z}_2 \times \mathbb{Z}_4$. The $\mathbb{Z}_2$ factor is generated by the element $\left(0, 0, \frac{1}{2}, \frac{1}{2}\right) \in Z_{F,\mathcal{P}_1} \oplus Z_{F,\mathcal{P}_2} \oplus Z_{F,\mathcal{P}_3} \simeq \mathbb{Z}_2 \times \mathbb{R}/\mathbb{Z} \times \mathbb{Z}_2 \times \mathbb{Z}_4$.

We can confirm this result using the analysis of Section 2.2 since the resulting $4d$ $\mathcal{N} = 2$ theory admits a Lagrangian description. The $4d$ theory is a bunch of free hypers transforming as

$$\begin{array}{c} \left[\mathfrak{u}(1)\right] \\ \Big| \mathsf{F} \\ \left[\mathfrak{su}(2)_2\right] \xrightarrow{\mathsf{F}} \quad \Big| \quad \xrightarrow{\mathsf{F}} \left[\mathfrak{su}(4)\right] \xrightarrow{\Lambda^2 \quad \frac{1}{2}\mathsf{F}} \left[\mathfrak{su}(2)_1\right] \end{array}, \tag{32}$$

where half-hypers are denoted by inserting a $\frac{1}{2}$ in front of one of the reps that the half-hyper transforms in. In total, we have a full-hyper transforming in $\mathsf{F} \otimes \mathsf{F} \otimes \mathbf{1}_1$ representation of $\mathfrak{su}(2)_2 \oplus \mathfrak{su}(4) \oplus \mathfrak{u}(1)$ and a half-hyper transforming in $\Lambda^2 \otimes \mathsf{F}$ representation of $\mathfrak{su}(4) \oplus \mathfrak{su}(2)_1$ where $\Lambda^2$ denotes the 2-index antisymmetric irrep of $\mathfrak{su}(4)$. For this Lagrangian description, we have $\mathcal{M} = \widetilde{Y}_F$ as one can notice that the two hypers correspond to the two generators $(1,0,0,2)$ and $(0,1,1,1)$ of $\widetilde{Y}_F$. Thus we are lead to precisely the same result (31).

$D_{2n}$ **example – Appearance of reducible $\mathcal{R}_o^\vee$ :**

$$\mathfrak{g} = \mathfrak{so}(8)$$

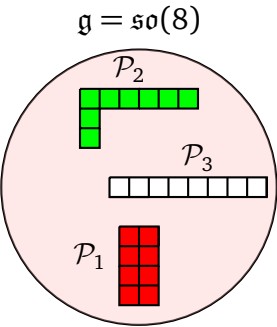

The puncture $\mathcal{P}_3$ has an $\mathfrak{f}_{\mathcal{P}_3} = \mathfrak{so}(8)$ flavor symmetry, $\mathcal{P}_2$ has $\mathfrak{f}_{\mathcal{P}_2} = \mathfrak{sp}(2)$, and $\mathcal{P}_1$ has $\mathfrak{f}_{\mathcal{P}_1} = \mathfrak{su}(2)$. The representation $\mathcal{R}_o^\vee = \mathsf{S} \oplus \mathsf{C}$ is the direct sum of spinor and cospinor irreps of $\mathfrak{h}_o^\vee = \mathfrak{g} = \mathfrak{so}(8)$. We have $\mathcal{R}_{o,s,\mathcal{P}_3}^\vee = \mathsf{S}$ and $\mathcal{R}_{o,c,\mathcal{P}_3} = \mathsf{C}$, $\mathcal{R}_{o,s,\mathcal{P}_2}^\vee = 2 \cdot \mathsf{F}$ and $\mathcal{R}_{o,c,\mathcal{P}_2}^\vee = 2 \cdot \mathsf{F}$ with $\mathsf{F}$ being the fundamental representation of $\mathfrak{f}_{\mathcal{P}_2} = \mathfrak{sp}(2)$, and $\mathcal{R}_{o,s,\mathcal{P}_1}^\vee = \mathsf{A} \oplus 5 \cdot \mathbf{1}$ and $\mathcal{R}_{o,c,\mathcal{P}_1}^\vee = 4 \cdot \mathsf{F}$ where $\mathsf{A}$ is the adjoint of $\mathfrak{f}_{\mathcal{P}_1} = \mathfrak{su}(2)$. From this we compute that $Y_{F,\mathcal{P}_i} = 0$ for all $i$. Thus, $Y_F' = 0$.

The local operators in representation (21) imply that $\widetilde{Y}_F$ is generated by the elements $(0,1,1,0)$ and $(1,0,1,1)$ in $\widehat{Z}_{F,\mathcal{P}_1} \oplus \widehat{Z}_{F,\mathcal{P}_2} \oplus \widehat{Z}_{F,\mathcal{P}_3} \simeq \mathbb{Z}/2\mathbb{Z} \oplus \mathbb{Z}/2\mathbb{Z} \oplus (\mathbb{Z}/2\mathbb{Z})_s \oplus (\mathbb{Z}/2\mathbb{Z})_c$. Since $Y_F' = 0$, we have $Y_F = \widetilde{Y}_F$ using which we compute that the manifest flavor symmetry group is

$$\mathcal{F} = \frac{SU(2) \times Sp(2) \times Spin(8)}{\mathbb{Z}_2 \times \mathbb{Z}_2}, \tag{33}$$

where one $\mathbb{Z}_2$ factor is generated by the element $\left(\frac{1}{2}, 0, 0, \frac{1}{2}\right) \in Z_{F,\mathcal{P}_1} \oplus Z_{F,\mathcal{P}_2} \oplus Z_{F,\mathcal{P}_3} \simeq \mathbb{Z}_2 \times \mathbb{Z}_2 \times (\mathbb{Z}_2)_s \times (\mathbb{Z}_2)_c$. The other $\mathbb{Z}_2$ factor is generated by the element $\left(0, \frac{1}{2}, \frac{1}{2}, \frac{1}{2}\right) \in Z_{F,\mathcal{P}_1} \oplus Z_{F,\mathcal{P}_2} \oplus Z_{F,\mathcal{P}_3} \simeq \mathbb{Z}_2 \times \mathbb{Z}_2 \times (\mathbb{Z}_2)_s \times (\mathbb{Z}_2)_c$.

We can confirm this result using the analysis of Section 2.2 since the resulting $4d$ $\mathcal{N} = 2$ theory admits a Lagrangian description. The $4d$ theory is a bunch of free hypers transforming as

$$\left[\mathfrak{sp}(2)\right] \xrightarrow{\frac{1}{2}\mathsf{F} \qquad \mathsf{S}} \left[\mathfrak{so}(8)\right] \xrightarrow{\mathsf{F} \qquad \frac{1}{2}\mathsf{F}} \left[\mathfrak{su}(2)\right], \tag{34}$$

where any the representation $\mathsf{F}$ of $\mathfrak{so}(8)$ is the vector irrep. For this Lagrangian description, we have $\mathcal{M} = \widetilde{Y}_F$ and we are lead to precisely the same result (33).

$D_{2n+1}$ **example – All $\mathcal{R}_o^\vee$ are irreducible in contrast with $D_{2n}$ :**

$$\mathfrak{g} = \mathfrak{so}(10)$$

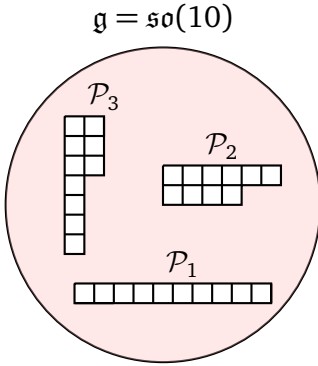

The puncture $\mathcal{P}_1$ has an $\mathfrak{f}_{\mathcal{P}_1} = \mathfrak{so}(10)$ flavor symmetry, $\mathcal{P}_2$ has $\mathfrak{f}_{\mathcal{P}_2} = \mathfrak{sp}(2) \oplus \mathfrak{u}(1)$, and $\mathcal{P}_3$ has a trivial flavor symmetry. The representation $\mathcal{R}_o^\vee$ is the spinor irrep $\mathsf{S}$ of $\mathfrak{h}_o^\vee = \mathfrak{g} = \mathfrak{so}(10)$. We

have $\mathcal{R}_{o,\mathcal{P}_1}^{\vee} = \mathsf{S}$ of $\mathfrak{f}_{\mathcal{P}_1} = \mathfrak{so}(10)$, $\mathcal{R}_{o,\mathcal{P}_2}^{\vee} = \Lambda_1^2 + 2 \cdot \mathsf{F}_{-1} + 3 \cdot \mathbf{1}_1$ of $\mathfrak{f}_{\mathcal{P}_2} = \mathfrak{sp}(2) \oplus \mathfrak{u}(1)$ where $\Lambda^2$ is the 2-index antisymmetric irrep of $\mathfrak{sp}(2)$ and subscripts specify the $\mathfrak{u}(1)$ charges. From this we compute that $Y_{F,\mathcal{P}_1} = 0$ and $Y_{F,\mathcal{P}_2}$ is the sub-lattice generated by $(1,2)$ in $\widehat{Z}_{F,\mathcal{P}_2} = \mathbb{Z}/2\mathbb{Z} \oplus \mathbb{Z}$. Thus, $Y_F'$ is generated by the element $(0,1,2) \in \widehat{Z}_{F,\mathcal{P}_1} \oplus \widehat{Z}_{F,\mathcal{P}_2} \simeq \mathbb{Z}/4\mathbb{Z} \oplus \mathbb{Z}/2\mathbb{Z} \oplus \mathbb{Z}$.

The local operators in representation (20) imply that $\widetilde{Y}_F$ is generated by the elements $(1,0,1)$ in $\widehat{Z}_{F,\mathcal{P}_1} \oplus \widehat{Z}_{F,\mathcal{P}_2}$. From this, we compute that the manifest flavor symmetry group is

$$\mathcal{F} = \frac{Spin(10) \times Sp(2) \times U(1)}{\mathbb{Z}_4}, \tag{35}$$

where $\mathbb{Z}_4$ is generated by the element $\left(\frac{1}{4}, \frac{1}{2}, -\frac{1}{4}\right) \in Z_{F,\mathcal{P}_1} \oplus Z_{F,\mathcal{P}_2} \simeq \mathbb{Z}_4 \times \mathbb{Z}_2 \times \mathbb{R}/\mathbb{Z}$.

We can confirm this result using the analysis of Section 2.2 since the resulting $4d$ $\mathcal{N} = 2$ theory admits a Lagrangian description. The $4d$ theory is a bunch of free hypers transforming as

$$\left[\mathfrak{u}(1)\right] \xrightarrow{\quad \mathsf{F} \qquad \mathsf{S} \quad} \left[\mathfrak{so}(10)\right] \xrightarrow{\quad \mathsf{F} \qquad \frac{1}{2}\mathsf{F} \quad} \left[\mathfrak{sp}(2)\right]. \tag{36}$$

For this Lagrangian description, we can compute $\mathcal{M}$ to find that it is precisely the sub-lattice of $\widehat{Z}_F$ generated by the generators of $Y_F'$ and $\widetilde{Y}_F$ discussed above. Thus we are lead to precisely the same result (35).

**First twisted example :**

$$\mathfrak{g} = \mathfrak{su}(4)$$

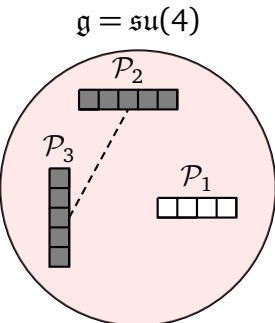

Consider compactifying $\mathfrak{g} = \mathfrak{su}(4)$ $6d$ $\mathcal{N} = (2,0)$ theory on a sphere with three regular punctures – one untwisted and maximal labeled $\mathcal{P}_1$, one *twisted* and maximal labeled $\mathcal{P}_2$, and one *twisted* and minimal labeled $\mathcal{P}_3$. The dashed line between the two twisted punctures is the open outer-automorphism twist line joining the two punctures.

The puncture $\mathcal{P}_1$ has an $\mathfrak{f}_{\mathcal{P}_1} = \mathfrak{su}(4)$ flavor symmetry, $\mathcal{P}_2$ has $\mathfrak{f}_{\mathcal{P}_2} = \mathfrak{sp}(2)$, and $\mathcal{P}_3$ has a trivial flavor symmetry. For $\mathcal{P}_1$ we have $\mathfrak{h}_o^{\vee} = \mathfrak{g} = \mathfrak{su}(4)$, $\mathcal{R}_o^{\vee} = \mathsf{F}$ of $\mathfrak{h}_o^{\vee} = \mathfrak{g} = \mathfrak{su}(4)$, and $\mathcal{R}_{o,\mathcal{P}_1}^{\vee} = \mathsf{F}$ of $\mathfrak{f}_{\mathcal{P}_1} = \mathfrak{su}(4)$. For $\mathcal{P}_2$ we have $\mathfrak{h}_o^{\vee} = \mathfrak{sp}(2)$, $\mathcal{R}_o^{\vee} = \mathsf{F}$ of $\mathfrak{h}_o^{\vee} = \mathfrak{sp}(2)$, and $\mathcal{R}_{o,\mathcal{P}_2}^{\vee} = \mathsf{F}$ of $\mathfrak{f}_{\mathcal{P}_2} = \mathfrak{sp}(2)$. From this we compute that $Y_{F,\mathcal{P}_1} = Y_{F,\mathcal{P}_2} = 0$ and thus $Y_F' = 0$.

All twist lines are of the same type and they are associated to $\mathfrak{h}_o^{\vee} = \mathfrak{sp}(2)$ which has corresponding $\mathcal{R}_o^{\vee} = \mathsf{F}$ of $\mathfrak{h}_o^{\vee} = \mathfrak{sp}(2)$ and $\mathsf{R}_o = \Lambda^2$ of $\mathfrak{g} = \mathfrak{su}(4)$. The local operators in representation (22) imply that $\widetilde{Y}_F$ is generated by the element $(2,1)$ in $\widehat{Z}_{F,\mathcal{P}_1} \oplus \widehat{Z}_{F,\mathcal{P}_2} \simeq \mathbb{Z}/4\mathbb{Z} \oplus \mathbb{Z}/2\mathbb{Z}$. From this, we compute that the manifest flavor symmetry group is

$$\mathcal{F} = \frac{SU(4) \times Sp(2)}{\mathbb{Z}_4}, \tag{37}$$

where $\mathbb{Z}_4$ is generated by the element $\left(\frac{1}{4}, \frac{1}{2}\right) \in Z_{F,\mathcal{P}_1} \oplus Z_{F,\mathcal{P}_2} \simeq \mathbb{Z}_4 \times \mathbb{Z}_2$.

We can confirm this result using the analysis of Section 2.2 since the resulting $4d$ $\mathcal{N} = 2$ theory admits a Lagrangian description. The $4d$ theory is a bunch of free hypers transforming

as

$$\left[\mathfrak{su}(4)\right] \xrightarrow{\quad\Lambda^2 \qquad \frac{1}{2}\mathsf{F}\quad} \left[\mathfrak{sp}(2)\right].$$

(38)

For this Lagrangian description, we can compute $\mathcal{M}$ to find that it is precisely equal to $\widetilde{Z}_F$ discussed above. Thus we are lead to precisely the same result (37).

**Example including non-minimal and non-maximal twisted punctures :**

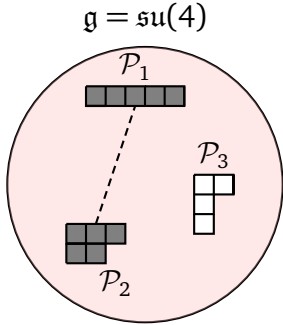

The puncture $\mathcal{P}_1$ has an $\mathfrak{f}_{\mathcal{P}_1} = \mathfrak{sp}(2)$ flavor symmetry, $\mathcal{P}_2$ has $\mathfrak{f}_{\mathcal{P}_2} = \mathfrak{su}(2)$, and $\mathcal{P}_3$ has $\mathfrak{f}_{\mathcal{P}_3} = \mathfrak{u}(1)$. For $\mathcal{P}_1$ we have $\mathfrak{h}_o^\vee = \mathfrak{sp}(2)$, $\mathcal{R}_o^\vee = \mathsf{F}$ of $\mathfrak{h}_o^\vee = \mathfrak{sp}(2)$, and $\mathcal{R}_{o,\mathcal{P}_1}^\vee = \mathsf{F}$ of $\mathfrak{f}_{\mathcal{P}_1} = \mathfrak{sp}(2)$. For $\mathcal{P}_2$ we have $\mathfrak{h}_o^\vee = \mathfrak{sp}(2)$, $\mathcal{R}_o^\vee = \mathsf{F}$ of $\mathfrak{h}_o^\vee = \mathfrak{sp}(2)$, and $\mathcal{R}_{o,\mathcal{P}_2}^\vee = \mathsf{F} \oplus 2 \cdot \mathbf{1}$ of $\mathfrak{f}_{\mathcal{P}_2} = \mathfrak{su}(2)$. For $\mathcal{P}_3$ we have $\mathfrak{h}_o^\vee = \mathfrak{g} = \mathfrak{su}(4)$, $\mathcal{R}_o^\vee = \mathsf{F}$ of $\mathfrak{h}_o^\vee = \mathfrak{g} = \mathfrak{su}(4)$, and $\mathcal{R}_{o,\mathcal{P}_3}^\vee = 3 \cdot \mathbf{1}_1 \oplus \mathbf{1}_{-3}$ of $\mathfrak{f}_{\mathcal{P}_3} = \mathfrak{u}(1)$. From this we compute that $Y_{F,\mathcal{P}_1} = 0$, $Y_{F,\mathcal{P}_2} = \widehat{Z}_{F,\mathcal{P}_2}$ and $Y_{F,\mathcal{P}_3} = 4\mathbb{Z}$ if we represent $\widehat{Z}_{F,\mathcal{P}_3} = \mathbb{Z}$.

All twist lines are of the same type and they are associated to $\mathfrak{h}_o^\vee = \mathfrak{sp}(2)$ which has corresponding $\mathcal{R}_o^\vee = \mathsf{F}$ of $\mathfrak{h}_o^\vee = \mathfrak{sp}(2)$ and $R_o = \Lambda^2$ of $\mathfrak{g} = \mathfrak{su}(4)$. The representation (22) becomes $\mathcal{R} = \mathsf{F} \otimes (\mathsf{F} \oplus 2 \cdot \mathbf{1}) \otimes (3 \cdot \mathbf{1}_{-2} \oplus 3 \cdot \mathbf{1}_2)$ of $\mathfrak{f}_{\mathcal{P}_1} \oplus \mathfrak{f}_{\mathcal{P}_2} \oplus \mathfrak{f}_{\mathcal{P}_3}$. From this, we conclude that the full $Y_F$ is generated by the elements $(0,1,0)$ and $(0,1,2)$ in $\widehat{Z}_{F,\mathcal{P}_1} \oplus \widehat{Z}_{F,\mathcal{P}_2} \oplus \widehat{Z}_{F,\mathcal{P}_3} \simeq \mathbb{Z}/2\mathbb{Z} \oplus \mathbb{Z}/2\mathbb{Z} \oplus \mathbb{Z}$. The manifest flavor symmetry group is then

$$\mathcal{F} = SU(2) \times \frac{Sp(2) \times U(1)}{\mathbb{Z}_4} = SU(2) \times \frac{Sp(2) \times U(1)/\mathbb{Z}_2}{\mathbb{Z}_2} \simeq SU(2) \times \frac{Sp(2) \times U(1)}{\mathbb{Z}_2}, \quad (39)$$

where the $\mathbb{Z}_4$ in $SU(2) \times \frac{Sp(2) \times U(1)}{\mathbb{Z}_4}$ is generated by the element $\left(\frac{1}{2}, 0, \frac{1}{4}\right) \in Z_{F,\mathcal{P}_1} \oplus Z_{F,\mathcal{P}_2} \oplus Z_{F,\mathcal{P}_3} \simeq \mathbb{Z}_2 \times \mathbb{Z}_2 \times \mathbb{R}/\mathbb{Z}$. The square of this element only acts on the $U(1)$ part, which allows us to write $\mathcal{F}$ as $SU(2) \times \frac{Sp(2) \times U(1)/\mathbb{Z}_2}{\mathbb{Z}_2}$. Redefining $U(1)/\mathbb{Z}_2 \simeq U(1)$ we obtain $\mathcal{F} = SU(2) \times \frac{Sp(2) \times U(1)}{\mathbb{Z}_2}$.

We can confirm this result using the analysis of Section 2.2 since the resulting 4$d$ $\mathcal{N} = 2$ theory admits a Lagrangian description. The 4$d$ theory is a bunch of free hypers transforming as

$$\left[\mathfrak{su}(2)\right] \xrightarrow{\quad\frac{1}{2}\mathsf{F} \qquad \Lambda^2\quad} \left[\mathfrak{sp}(2)\right] \xrightarrow{\quad\mathsf{F} \qquad \mathsf{F}\quad} \left[\mathfrak{u}(1)\right].$$

(40)

From this Lagrangian description, we can compute

$$\mathcal{F} = SU(2) \times \frac{Sp(2) \times U(1)}{\mathbb{Z}_2}, \quad (41)$$

which matches (39).

**Half-hyper in bifundamental; Twisted $D_{2n+1}$ example :**

Consider compactifying $\mathfrak{g} = \mathfrak{so}(4n+2)$ $6d$ $\mathcal{N} = (2,0)$ theory on a sphere with three regular punctures – one untwisted and maximal labeled $\mathcal{P}_1$, one twisted and maximal labeled $\mathcal{P}_2$, and one twisted and minimal labeled $\mathcal{P}_3$. The puncture $\mathcal{P}_1$ has an $\mathfrak{f}_{\mathcal{P}_1} = \mathfrak{so}(4n+2)$ flavor symmetry, $\mathcal{P}_2$ has $\mathfrak{f}_{\mathcal{P}_2} = \mathfrak{sp}(2n)$, and $\mathcal{P}_3$ has a trivial flavor symmetry. For $\mathcal{P}_1$ we have $\mathfrak{h}_o^\vee = \mathfrak{g} = \mathfrak{so}(4n+2)$, $\mathcal{R}_o^\vee = \mathsf{S}$ of $\mathfrak{h}_o^\vee = \mathfrak{g} = \mathfrak{so}(4n+2)$, and $\mathcal{R}_{o,\mathcal{P}_1}^\vee = \mathsf{S}$ of $\mathfrak{f}_{\mathcal{P}_1} = \mathfrak{so}(4n+2)$. For $\mathcal{P}_2$ we have $\mathfrak{h}_o^\vee = \mathfrak{sp}(2n)$, $\mathcal{R}_o^\vee = \mathsf{F}$ of $\mathfrak{h}_o^\vee = \mathfrak{sp}(2n)$, and $\mathcal{R}_{o,\mathcal{P}_2}^\vee = \mathsf{F}$ of $\mathfrak{f}_{\mathcal{P}_2} = \mathfrak{sp}(2n)$. From this we compute that $Y_{F,\mathcal{P}_1} = Y_{F,\mathcal{P}_2} = 0$ and thus $Y_F' = 0$.

All twist lines are of the same type and they are associated to $\mathfrak{h}_o^\vee = \mathfrak{sp}(2n)$ which has corresponding $\mathcal{R}_o^\vee = \mathsf{F}$ of $\mathfrak{h}_o^\vee = \mathfrak{sp}(2n)$ and $\mathsf{R}_o = \mathsf{F}$ which is the vector irrep of $\mathfrak{g} = \mathfrak{so}(4n+2)$. The local operators in representation (22) imply that $\widetilde{Y}_F$ is generated by the element $(2,1)$ in $\widehat{Z}_{F,\mathcal{P}_1} \oplus \widehat{Z}_{F,\mathcal{P}_2} \simeq \mathbb{Z}/4\mathbb{Z} \oplus \mathbb{Z}/2\mathbb{Z}$. From this, we compute that the manifest flavor symmetry group is

$$\mathcal{F} = \frac{Spin(4n+2) \times Sp(2n)}{\mathbb{Z}_4} = \frac{Spin(4n+2)/\mathbb{Z}_2 \times Sp(2n)}{\mathbb{Z}_2} = \frac{SO(4n+2) \times Sp(2n)}{\mathbb{Z}_2}, \quad (42)$$

where $\mathbb{Z}_4$ in $\frac{Spin(4n+2) \times Sp(2n)}{\mathbb{Z}_4}$ is generated by the element $\left(\frac{1}{4}, \frac{1}{2}\right) \in Z_{F,\mathcal{P}_1} \oplus Z_{F,\mathcal{P}_2} \simeq \mathbb{Z}_4 \times \mathbb{Z}_2$.

We can confirm this result using the analysis of Section 2.2 since the resulting $4d$ $\mathcal{N} = 2$ theory admits a Lagrangian description. The $4d$ theory is a bunch of free hypers transforming as

$$\left[\mathfrak{so}(4n+2)\right] \xrightarrow{\quad \mathsf{F} \qquad \frac{1}{2}\mathsf{F} \quad} \left[\mathfrak{sp}(2n)\right] . \quad (43)$$

For this Lagrangian description, we can compute $\mathcal{M}$ to find that it is precisely equal to $\widetilde{Z}_F$ discussed above. Thus we are lead to precisely the same result (42).

The case of $\mathfrak{g} = \mathfrak{so}(4n)$ with same punctures can be performed easily in a similar fashion as above and we obtain

$$\mathcal{F} = \frac{SO(4n) \times Sp(2n-1)}{\mathbb{Z}_2} . \quad (44)$$

**Twisted $D_{2n}$ example :**

$$\mathfrak{g} = \mathfrak{so}(8)$$

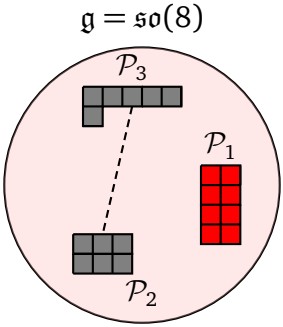

The puncture $\mathcal{P}_1$ has an $\mathfrak{f}_{\mathcal{P}_1} = \mathfrak{su}(2)_1$ flavor symmetry, $\mathcal{P}_2$ has $\mathfrak{f}_{\mathcal{P}_2} = \mathfrak{su}(2)_2$, and $\mathcal{P}_3$ has $\mathfrak{f}_{\mathcal{P}_3} = \mathfrak{sp}(2)$. For $\mathcal{P}_1$ we have $\mathfrak{h}_o^\vee = \mathfrak{g} = \mathfrak{so}(8)$, $\mathcal{R}_{o,s}^\vee = \mathsf{S}$ and $\mathcal{R}_{o,c}^\vee = \mathsf{C}$ of $\mathfrak{h}_o^\vee = \mathfrak{g} = \mathfrak{so}(8)$, and $\mathcal{R}_{o,s,\mathcal{P}_1}^\vee = \mathsf{A} \oplus 5 \cdot \mathbf{1}$ and $\mathcal{R}_{o,c,\mathcal{P}_1}^\vee = 4 \cdot \mathsf{F}$ of $\mathfrak{f}_{\mathcal{P}_1} = \mathfrak{su}(2)_1$. For $\mathcal{P}_2$ we have $\mathfrak{h}_o^\vee = \mathfrak{sp}(3)$, $\mathcal{R}_o^\vee = \mathsf{F}$ of $\mathfrak{h}_o^\vee = \mathfrak{sp}(3)$, and $\mathcal{R}_{o,\mathcal{P}_2}^\vee = 2 \cdot \mathsf{A}$ of $\mathfrak{f}_{\mathcal{P}_2} = \mathfrak{su}(2)_2$. For $\mathcal{P}_3$ we have $\mathfrak{h}_o^\vee = \mathfrak{sp}(3)$, and $\mathcal{R}_o^\vee = \mathsf{F}$ of $\mathfrak{h}_o^\vee = \mathfrak{sp}(3)$, and $\mathcal{R}_{o,\mathcal{P}_3}^\vee = \mathsf{F} \oplus 2 \cdot \mathbf{1}$ of $\mathfrak{f}_{\mathcal{P}_3} = \mathfrak{sp}(2)$. From this we compute that $Y_{F,\mathcal{P}_1} = Y_{F,\mathcal{P}_2} = 0$ and $Y_{F,\mathcal{P}_3} = \widehat{Z}_{F,\mathcal{P}_2}$.

All twist lines are of the same type and they are associated to $\mathfrak{h}_o^\vee = \mathfrak{sp}(3)$ which has corresponding $\mathcal{R}_o^\vee = \mathsf{F}$ of $\mathfrak{h}_o^\vee = \mathfrak{sp}(3)$ and $\mathsf{R}_o = \mathsf{F}$ of $\mathfrak{g} = \mathfrak{so}(8)$. The representation (22) becomes $\mathcal{R} = (4 \cdot \mathsf{F}) \otimes (2 \cdot \mathsf{A}) \otimes (\mathsf{F} \oplus 2 \cdot \mathbf{1})$ of $\mathfrak{f}_{\mathcal{P}_1} \oplus \mathfrak{f}_{\mathcal{P}_2} \oplus \mathfrak{f}_{\mathcal{P}_3}$. From this, we conclude that the full $Y_F$ is

generated by the elements $(0, 0, 1)$ and $(1, 0, 0)$ in $\widehat{Z}_{F,\mathcal{P}_1} \oplus \widehat{Z}_{F,\mathcal{P}_2} \oplus \widehat{Z}_{F,\mathcal{P}_3} \simeq \mathbb{Z}/2\mathbb{Z} \oplus \mathbb{Z}/2\mathbb{Z} \oplus \mathbb{Z}/2\mathbb{Z}$. The manifest flavor symmetry group is then

$$\mathcal{F} = SU(2)_1 \times SO(3)_2 \times Sp(2). \tag{45}$$

Note that by moving the puncture  through the $\mathbb{Z}_2$ outer-automorphism twist line, it is

converted into ![blue puncture]. So the "color" of the puncture should not influence the flavor symmetry group $\mathcal{F}$. Indeed this is the case and is in-built into the way the calculation of $\mathcal{F}$ is performed.

We can confirm the above result (45) using the analysis of Section 2.2 since the resulting $4d$ $\mathcal{N} = 2$ theory admits a Lagrangian description. The $4d$ theory is a bunch of free hypers transforming as

$$
\begin{array}{c}
\left[\mathfrak{su}(2)_1\right] \xrightarrow[\frac{1}{2}\mathsf{F}\quad\quad\Lambda^2]{} \left[\mathfrak{sp}(2)\right] \\[4pt]
{}^{\frac{1}{2}\mathsf{F}} \diagdown\quad\quad \big|\, {}^{\frac{1}{2}\mathsf{F}} \\[8pt]
\mathsf{A} \diagdown\quad \big|\, \mathsf{A} \\[4pt]
\left[\mathfrak{su}(2)_2\right]
\end{array}
\tag{46}
$$

From this Lagrangian description, we can verify (45).

$E_6$ **example :** Consider compactifying $\mathfrak{g} = \mathfrak{e}_6$ $6d$ $\mathcal{N} = (2, 0)$ theory on a sphere with three untwisted regular punctures – one maximal specified by Nahm Bala-Carter (BC) label $0$ and labeled $\mathcal{P}_1$, one specified by Nahm BC label $A_2 + 2A_1$ and labeled $\mathcal{P}_2$, and one specified by Nahm BC label $E_6(a_1)$ and labeled $\mathcal{P}_3$. The puncture $\mathcal{P}_1$ has an $\mathfrak{f}_{\mathcal{P}_1} = \mathfrak{e}_6$ flavor symmetry, $\mathcal{P}_2$ has $\mathfrak{f}_{\mathcal{P}_2} = \mathfrak{su}(2) \oplus \mathfrak{u}(1)$, and $\mathcal{P}_3$ has a trivial flavor symmetry. The representation $\mathcal{R}_o^\vee$ is the 27-dimensional irrep $\mathsf{F}$ of $\mathfrak{h}_o^\vee = \mathfrak{g} = \mathfrak{e}_6$. We have $\mathcal{R}_{o,\mathcal{P}_1}^\vee = \mathsf{F}$ of $\mathfrak{f}_{\mathcal{P}_1} = \mathfrak{e}_6$, $\mathcal{R}_{o,\mathcal{P}_2}^\vee = \mathbf{1}_2 \oplus \mathbf{1}_{-4} \oplus 2 \cdot \mathbf{4}_{-1} \oplus 3 \cdot \mathbf{3}_2 \oplus 4 \cdot \mathbf{2}_{-1}$ of $\mathfrak{f}_{\mathcal{P}_2} = \mathfrak{su}(2) \oplus \mathfrak{u}(1)$ where $\mathbf{n}$ denotes the $n$-dimensional irrep of $\mathfrak{su}(2)$ and subscripts specify the $\mathfrak{u}(1)$ charges. From this we compute that $Y_{F,\mathcal{P}_1} = 0$ and $Y_{F,\mathcal{P}_2}$ is the sub-lattice generated by $(1, 3)$ in $\widehat{Z}_{F,\mathcal{P}_2} = \mathbb{Z}/2\mathbb{Z} \oplus \mathbb{Z}$. Thus, $Y_F'$ is generated by the element $(0, 1, 3) \in \widehat{Z}_{F,\mathcal{P}_1} \oplus \widehat{Z}_{F,\mathcal{P}_2} \simeq \mathbb{Z}/3\mathbb{Z} \oplus \mathbb{Z}/2\mathbb{Z} \oplus \mathbb{Z}$.

Accounting for the local operators in representation (20), we find that the full $Y_F$ is generated by the element $(-1, 1, 1)$ in $\widehat{Z}_{F,\mathcal{P}_1} \oplus \widehat{Z}_{F,\mathcal{P}_2}$. From this, we compute that the manifest flavor symmetry group is

$$\mathcal{F} = \frac{E_6 \times SU(2) \times U(1)}{\mathbb{Z}_6}, \tag{47}$$

where $\mathbb{Z}_6$ is generated by the element $\left(\frac{1}{3}, \frac{1}{2}, \frac{1}{6}\right) \in Z_{F,\mathcal{P}_1} \oplus Z_{F,\mathcal{P}_2} \simeq \mathbb{Z}_3 \times \mathbb{Z}_2 \times \mathbb{R}/\mathbb{Z}$.

We can confirm this result using the analysis of Section 2.2 since the resulting $4d$ $\mathcal{N} = 2$ theory admits a Lagrangian description. The $4d$ theory is a bunch of free hypers transforming as

$$
\begin{array}{c}
\left[\mathfrak{u}(1)\right] \\[4pt]
\big|\, \mathsf{F} \\[8pt]
\left[\mathfrak{e}_6\right] \xrightarrow[\mathsf{F}\quad\quad\quad\mathsf{F}]{} \left[\mathfrak{su}(2)\right]
\end{array}
\tag{48}
$$

From this Lagrangian description, we can easily confirm (47).

$E_7$ **example :** Consider compactifying $\mathfrak{g} = \mathfrak{e}_7$ $6d$ $\mathcal{N} = (2,0)$ theory on a sphere with three untwisted regular punctures – one maximal specified by Nahm Bala-Carter (BC) label 0 and labeled $\mathcal{P}_1$, one specified by Nahm BC label $A_3 + A_2 + A_1$ and labeled $\mathcal{P}_2$, and one specified by Nahm BC label $E_7(a_1)$ and labeled $\mathcal{P}_3$. The puncture $\mathcal{P}_1$ has an $\mathfrak{f}_{\mathcal{P}_1} = \mathfrak{e}_7$ flavor symmetry, $\mathcal{P}_2$ has $\mathfrak{f}_{\mathcal{P}_2} = \mathfrak{su}(2)$, and $\mathcal{P}_3$ has a trivial flavor symmetry. The representation $\mathcal{R}_o^\vee$ is the 56-dimensional irrep F of $\mathfrak{h}_o^\vee = \mathfrak{g} = \mathfrak{e}_7$. We have $\mathcal{R}_{o,\mathcal{P}_1}^\vee = \mathsf{F}$ of $\mathfrak{f}_{\mathcal{P}_1} = \mathfrak{e}_7$, $\mathcal{R}_{o,\mathcal{P}_2}^\vee = 2 \cdot \mathbf{5} \oplus 4 \cdot \mathbf{7} \oplus 6 \cdot \mathbf{3}$ of $\mathfrak{f}_{\mathcal{P}_2} = \mathfrak{su}(2)$ where $\mathbf{n}$ denotes the $n$-dimensional irrep of $\mathfrak{su}(2)$. From this we compute that $Y_{F,\mathcal{P}_1} = Y_{F,\mathcal{P}_2} = 0$.

Accounting for the local operators in representation (20), we find that the full $Y_F$ is generated by the element $(1,0)$ in $\widehat{Z}_{F,\mathcal{P}_1} \oplus \widehat{Z}_{F,\mathcal{P}_2} \simeq \mathbb{Z}/2\mathbb{Z} \oplus \mathbb{Z}/2\mathbb{Z}$. From this, we compute that the manifest flavor symmetry group is

$$\mathcal{F} = E_7 \times SO(3). \tag{49}$$

We can confirm this result using the analysis of Section 2.2 since the resulting $4d$ $\mathcal{N} = 2$ theory admits a Lagrangian description. The $4d$ theory is a bunch of free hypers transforming as

$$\left[\mathfrak{e}_7\right] \xrightarrow{\frac{1}{2}\mathsf{F} \qquad A} \left[\mathfrak{su}(2)\right]. \tag{50}$$

From this Lagrangian description, we can easily confirm (49).

## 4.2 Theories Having Weakly-coupled Duality Frames

$SU(n) + 2n\mathsf{F}$ **SCFT :** Consider compactifying $\mathfrak{g} = \mathfrak{su}(n)$ $6d$ $\mathcal{N} = (2,0)$ theory on a sphere with four untwisted regular punctures – two maximal labeled $\mathcal{P}_1, \mathcal{P}_2$ and two minimal labeled $\mathcal{P}_3, \mathcal{P}_4$. The punctures $\mathcal{P}_1, \mathcal{P}_2$ have $\mathfrak{f}_{\mathcal{P}_i} = \mathfrak{su}(n)_i$, and $\mathcal{P}_3, \mathcal{P}_4$ have $\mathfrak{f}_{\mathcal{P}_i} = \mathfrak{u}(1)_i$. The representation $\mathcal{R}_o^\vee$ is the fundamental representation of $\mathfrak{h}_o^\vee = \mathfrak{g} = \mathfrak{su}(n)$ and thus $\mathcal{R}_{o,\mathcal{P}_i}^\vee$ is the fundamental representation $\mathsf{F}_i$ of $\mathfrak{f}_{\mathcal{P}_i} = \mathfrak{su}(n)_i$ for $i = 1, 2$, and $\mathcal{R}_{o,\mathcal{P}_i}^\vee = (n-1) \cdot \mathbf{1}_1 \oplus \mathbf{1}_{1-n}$ of $\mathfrak{f}_{\mathcal{P}_i} = \mathfrak{u}(1)_i$ for $i = 3, 4$. From this we compute that $Y_{F,\mathcal{P}_i} = 0$ for $i = 1, 2$, but $Y_{F,\mathcal{P}_i} = n\mathbb{Z}$ for $i = 3, 4$ if we represent $\widehat{Z}_{F,\mathcal{P}_i} = \mathbb{Z}$.

The representation (20) becomes

$$\mathsf{F}_1 \otimes \mathsf{F}_2 \otimes \left((n-1) \cdot \mathbf{1}_1 \oplus \mathbf{1}_{1-n}\right) \otimes \left((n-1) \cdot \mathbf{1}_1 \oplus \mathbf{1}_{1-n}\right), \tag{51}$$

implying that $\widetilde{Y}_F$ is generated by the element $(1,1,1,1) \in \widehat{Z}_{F,\mathcal{P}_1} \oplus \widehat{Z}_{F,\mathcal{P}_2} \oplus \widehat{Z}_{F,\mathcal{P}_3} \oplus \widehat{Z}_{F,\mathcal{P}_4}$. From this we compute that the manifest flavor symmetry group is

$$\mathcal{F} = \frac{SU(n)_1 \times SU(n)_2 \times U(1)_3 \times U(1)_4}{\mathbb{Z}_n^{1,2} \times \mathbb{Z}_n^{2,3} \times \mathbb{Z}_n^{3,4}}, \tag{52}$$

where $\mathbb{Z}_n^{1,2}$ is the $\mathbb{Z}_n$ subgroup generated by the element $\left(\frac{1}{n}, -\frac{1}{n}, 0, 0\right)$ of the $(\mathbb{Z}_n)_1 \times (\mathbb{Z}_n)_2 \times (\mathbb{R}/\mathbb{Z})_3 \times (\mathbb{R}/\mathbb{Z})_4$ center. The $\mathbb{Z}_n^{2,3}$ factor is generated by the element $\left(0, \frac{1}{n}, -\frac{1}{n}, 0\right)$ and the $\mathbb{Z}_n^{3,4}$ factor is generated by the element $\left(0, 0, \frac{1}{n}, -\frac{1}{n}\right)$.

We can confirm this result using the analysis of Section 2.2 since the resulting $4d$ $\mathcal{N} = 2$ theory admits a duality frame with the following Lagrangian description

$$
\begin{array}{ccccc}
\left[\mathfrak{u}(1)_3\right] & & \left[\mathfrak{u}(1)_4\right] & \\
\Big| \mathsf{F} & & \Big| \mathsf{F} & \\
\left[\mathfrak{su}(n)_1\right] \xrightarrow{\ \mathsf{F}\ \ \ \ \ \ \ \ \mathsf{F}\ } \mathfrak{su}(n) \xrightarrow{\ \mathsf{F}\ \ \ \ \ \ \ \ \mathsf{F}\ } \left[\mathfrak{su}(n)_2\right] &
\end{array}, \tag{53}
$$

where an algebra not in brackets denotes a gauge algebra. For this Lagrangian description, we have $\mathcal{E} \simeq \mathbb{Z}_n^3$ generated by elements $\left(-\frac{1}{n}, \frac{1}{n}, -\frac{1}{n}, 0, 0\right)$, $\left(0, 0, \frac{1}{n}, -\frac{1}{n}, 0\right)$ and $\left(\frac{1}{n}, 0, 0, \frac{1}{n}, -\frac{1}{n}\right)$ in $Z_G \times Z_F \simeq \mathbb{Z}_n \times (\mathbb{Z}_n)_1 \times (\mathbb{Z}_n)_2 \times (\mathbb{R}/\mathbb{Z})_3 \times (\mathbb{R}/\mathbb{Z})_4$. Projecting $\mathcal{E}$ onto $Z_F$ we find precisely the subgroup $\mathcal{Z} \simeq \mathbb{Z}_n^{1,2} \times \mathbb{Z}_n^{2,3} \times \mathbb{Z}_n^{3,4} \subset Z_F$ and we are lead to precisely the same result (52).

**Example: Higher genus; Twisted $A_{2n}, A_{2n-1}$ for large enough $n$ :** Consider compactifying $\mathfrak{g} = \mathfrak{su}(n)$ $6d$ $\mathcal{N} = (2,0)$ theory on a torus with an untwisted minimal regular puncture labeled $\mathcal{P}$ and a closed $\mathbb{Z}_2$ twist line wrapping a non-trivial cycle of the torus. We have $\mathfrak{f}_{\mathcal{P}} = \mathfrak{u}(1)$. For the puncture $\mathcal{P}$, we have $\mathfrak{h}_o^\vee = \mathfrak{g} = \mathfrak{su}(n)$, $\mathcal{R}_o^\vee = \mathsf{F}$ of $\mathfrak{h}_o^\vee = \mathfrak{g} = \mathfrak{su}(n)$, and $\mathcal{R}_{o,\mathcal{P}}^\vee = (n-1) \cdot \mathbf{1}_1 \oplus \mathbf{1}_{1-n}$ of $\mathfrak{f}_{\mathcal{P}} = \mathfrak{u}(1)$. From this we compute that $Y_{F,\mathcal{P}} = n\mathbb{Z}$ if we represent $\widehat{Z}_{F,\mathcal{P}} = \mathbb{Z}$.

For further analysis we need to distinguish whether $n$ is even or odd. Let us first consider $n = 2m$. All twist lines are of the same type and they are associated to $\mathfrak{h}_o^\vee = \mathfrak{so}(2m+1)$ which has corresponding $\mathcal{R}_o^\vee = \mathsf{S}$ of $\mathfrak{h}_o^\vee = \mathfrak{so}(2m+1)$ and $\mathsf{R}_o = \Lambda^m$ i.e. the $m$-index antisymmetric irrep of $\mathfrak{g} = \mathfrak{su}(2m)$. The representation (22) becomes $\mathcal{R} = \frac{1}{2}\binom{2m}{m} \cdot \mathbf{1}_m \oplus \frac{1}{2}\binom{2m}{m} \cdot \mathbf{1}_{-m}$ of $\mathfrak{f}_{\mathcal{P}} = \mathfrak{u}(1)$. From this, we find that the full $Y_F = m\mathbb{Z}$ if we represent $\widehat{Z}_F = \widehat{Z}_{F,\mathcal{P}} = \mathbb{Z}$. The manifest flavor symmetry group is then

$$\mathcal{F} = \frac{U(1)}{\mathbb{Z}_m}, \tag{54}$$

where $U(1)$ appearing in the numerator is the global form of $\mathfrak{f}_{\mathcal{P}} = \mathfrak{u}(1)$ for which various charges above were listed. This $U(1)$ will also appear naturally in the gauge theory description that we will study below.

Now consider $n = 2m + 1$. The twist lines are now associated to $\mathfrak{h}_o^\vee = \mathfrak{sp}(m)$ which has corresponding $\mathcal{R}_o^\vee = \mathsf{F}$ of $\mathfrak{h}_o^\vee = \mathfrak{sp}(m)$ and $\mathsf{R}_o = \mathsf{A}$ of $\mathfrak{g} = \mathfrak{su}(2m+1)$. The representation (22) becomes $\mathcal{R} = 2m \cdot \mathbf{1}_{2m+1} \oplus 2m \cdot \mathbf{1}_{-2m-1} \oplus (4m^2+1) \cdot \mathbf{1}_0$ of $\mathfrak{f}_{\mathcal{P}} = \mathfrak{u}(1)$. From this, we find that the full $Y_F = (2m+1)\mathbb{Z}$ if we represent $\widehat{Z}_F = \widehat{Z}_{F,\mathcal{P}} = \mathbb{Z}$. The manifest flavor symmetry group is then

$$\mathcal{F} = \frac{U(1)}{\mathbb{Z}_{2m+1}}, \tag{55}$$

where $U(1)$ appearing in the numerator is the global form of $\mathfrak{f}_{\mathcal{P}} = \mathfrak{u}(1)$ for which various charges above were listed. This $U(1)$ will also appear naturally in the gauge theory description that we will study below.

The above $4d$ $\mathcal{N} = 2$ theory admits a duality frame with the following Lagrangian description

$$\mathfrak{su}(n) \quad \overset{\mathsf{F}}{\underset{\mathsf{F}}{\diamondsuit}} \quad \overset{\mathsf{F}}{-} \big[\mathfrak{u}(1)\big] \quad , \tag{56}$$

i.e. the $\mathfrak{su}(n)$ gauge theory with a hyper transforming in $\mathsf{F} \otimes \mathsf{F}$ which is rotated by the flavor $\mathfrak{u}(1)$. For this Lagrangian description, we have $\mathcal{M}$ generated by the element $(2, 1) \in \widehat{Z}_G \times \widehat{Z}_F \simeq \mathbb{Z}/4\mathbb{Z} \times \mathbb{Z}$ where the charges under $Z_F$ are charges under the $U(1)$ appearing in the numerators of (54) and (55) [17]. From this we compute that $\mathcal{E}$ generated by the element $\left(-\frac{2}{n}, \frac{1}{n}\right) \in Z_G \times Z_F \simeq \mathbb{Z}_n \times \mathbb{R}/\mathbb{Z}$. For $n = 2m$, this leads to (54), and for $n = 2m + 1$, this leads to (55).

---

[17]This follows from the fact that the $U(1)$ appearing in the numerator of (29) is the same as the $U(1)$ under which the bifundamental hyper in (30) has charge $+1$. The theory (56) is obtained by gauging the diagonal $\mathfrak{su}(n)$ of the two $\mathfrak{su}(n)$ flavor symmetries present in (30) which corresponds to stitching the sphere along the maximal punctures into a torus with a closed outer-automorphism twist line wrapping the homologically non-trivial cycle created by the stitching. These operations do not touch the $U(1)$ symmetry appearing in (30) and the minimal puncture on the sphere, thus leaving the equality of the two $U(1)$s intact.

**Twisted $E_6$ example :**    Consider compactifying $\mathfrak{g} = \mathfrak{e}_6$ $6d$ $\mathcal{N} = (2,0)$ theory on a sphere with four regular punctures – one twisted with Nahm BC label $B_3$ and labeled $\mathcal{P}_1$, one untwisted with Nahm BC label $A_2 + 2A_1$ and labeled $\mathcal{P}_2$, one untwisted with Nahm BC label $E_6(a_1)$ and labeled $\mathcal{P}_3$, and one twisted with Nahm BC label $F_4$ and labeled $\mathcal{P}_4$. The puncture $\mathcal{P}_1$ has an $\mathfrak{f}_{\mathcal{P}_1} = \mathfrak{su}(2)_1$ flavor symmetry, $\mathcal{P}_2$ has $\mathfrak{f}_{\mathcal{P}_2} = \mathfrak{su}(2)_2 \oplus \mathfrak{u}(1)$, and $\mathcal{P}_3$ and $\mathcal{P}_4$ have a trivial flavor symmetry. For $\mathcal{P}_1$ we have $\mathfrak{h}_o^\vee = \mathfrak{f}_4$, $\mathcal{R}_o^\vee$ is the 26-dimensional irrep F of $\mathfrak{h}_o^\vee = \mathfrak{f}_4$, and $\mathcal{R}_{o,\mathcal{P}_1}^\vee = \mathbf{5} \oplus 7 \cdot \mathbf{3}$ of $\mathfrak{f}_{\mathcal{P}_1} = \mathfrak{su}(2)_1$. For $\mathcal{P}_2$ we have $\mathfrak{h}_o^\vee = \mathfrak{g} = \mathfrak{e}_6$, $\mathcal{R}_o^\vee = $ F of $\mathfrak{h}_o^\vee = \mathfrak{g} = \mathfrak{e}_6$, and $\mathcal{R}_{o,\mathcal{P}_2}^\vee = \mathbf{1}_2 \oplus \mathbf{1}_{-4} \oplus 2 \cdot \mathbf{4}_{-1} \oplus 3 \cdot \mathbf{3}_2 \oplus 4 \cdot \mathbf{2}_{-1}$ of $\mathfrak{f}_{\mathcal{P}_2} = \mathfrak{su}(2)_2 \oplus \mathfrak{u}(1)$. From this we compute that $Y_F'$ is generated by the element $(0, 1, 3) \in \widehat{Z}_{F,\mathcal{P}_1} \oplus \widehat{Z}_{F,\mathcal{P}_2} \simeq \mathbb{Z}/2\mathbb{Z} \oplus \mathbb{Z}/2\mathbb{Z} \oplus \mathbb{Z}$.

All twist lines are of the same type and they are associated to $\mathfrak{h}_o^\vee = \mathfrak{f}_4$ which has corresponding $\mathcal{R}_o^\vee = $ F of $\mathfrak{h}_o^\vee = \mathfrak{f}_4$ and $\mathsf{R}_o = $ A of $\mathfrak{g} = \mathfrak{e}_6$. The representation (22) becomes $\mathcal{R} = (\mathbf{5} \oplus 7 \cdot \mathbf{3}) \otimes (4 \cdot \mathbf{1}_0 \oplus 9 \cdot \mathbf{3}_0 \oplus 2 \cdot \mathbf{4}_3 \oplus 2 \cdot \mathbf{4}_{-3} \oplus 3 \cdot \mathbf{5}_0 \oplus 4 \cdot \mathbf{2}_3 \oplus 4 \cdot \mathbf{2}_{-3})$F of $\mathfrak{f}_{\mathcal{P}_1} \oplus \mathfrak{f}_{\mathcal{P}_2}$. From this, we conclude that the full $Y_F$ equals $Y_F'$. The manifest flavor symmetry group is then

$$
\begin{aligned}
\mathcal{F} = SO(3)_1 \times \frac{SU(2)_2 \times U(1)}{\mathbb{Z}_6} &= SO(3)_1 \times \frac{SU(2)_2 \times U(1)/\mathbb{Z}_3}{\mathbb{Z}_2} \\
&\simeq SO(3)_1 \times \frac{SU(2)_2 \times U(1)}{\mathbb{Z}_2} = SO(3)_1 \times U(2),
\end{aligned}
\tag{57}
$$

where the $\mathbb{Z}_6$ factor is generated by the element $\left(0, \frac{1}{2}, \frac{1}{6}\right) \in Z_{F,\mathcal{P}_1} \oplus Z_{F,\mathcal{P}_2} \simeq (\mathbb{Z}_2)_1 \times (\mathbb{Z}_2)_2 \times \mathbb{R}/\mathbb{Z}$ where $(\mathbb{Z}_2)_i$ is the center of $SU(2)_i$.

We can confirm this result using the analysis of Section 2.2 since the resulting $4d$ $\mathcal{N} = 2$ theory admits a duality frame with the following Lagrangian description

$$
\begin{array}{ccc}
 & & \big[\mathfrak{u}(1)\big] \\
 & & \Big| \, {}^{\text{F}} \\
\big[\mathfrak{su}(2)_1\big] \qquad \mathfrak{su}(6) & \xrightarrow{\Lambda^2 \oplus 2\bar{\text{F}}} & \xrightarrow{\ \text{F}\ } \big[\mathfrak{su}(2)_2\big]
\end{array}
\tag{58}
$$

where $\mathfrak{su}(2)_1$ does not act on the gauge theory. From this Lagrangian description, we see that $\mathcal{E}$ is generated by $\left(-\frac{1}{3}, 0, \frac{1}{2}, \frac{1}{6}\right) \in Z_G \times Z_F \simeq \mathbb{Z}_6 \times (\mathbb{Z}_2)_1 \times (\mathbb{Z}_2)_2 \times \mathbb{R}/\mathbb{Z}$. This leads us to the flavor symmetry group displayed in (57).

**$S_3$ Twisted $D_4$ example :**

$$\mathfrak{g} = \mathfrak{so}(8)$$

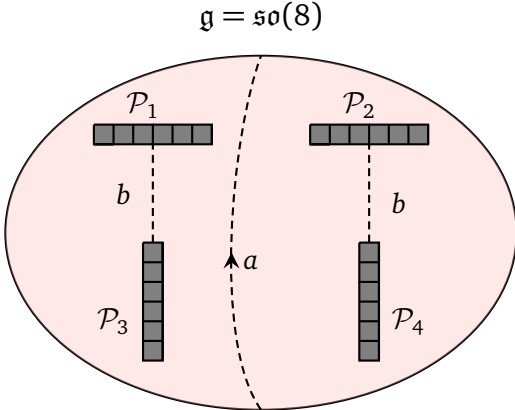

Here $a$ and $b$ denote two outer-automorphism elements in $\mathcal{O}_\mathfrak{g} = S_3$. $b$ is the element of order 2 whose action keeps the vector irrep of $\mathfrak{so}(8)$ invariant. $a$ is an order 3 element. The $a$ line is a closed twist line that wraps around the sphere as shown above.

The punctures $\mathcal{P}_i$ for $i = 1, 2$ have $\mathfrak{f}_{\mathcal{P}_i} = \mathfrak{sp}(3)_i$ flavor symmetry, and the punctures $\mathcal{P}_i$ for $i = 3, 4$ have trivial flavor symmetry. For every puncture we have $\mathfrak{h}_o^\vee = \mathfrak{sp}(3)$, $\mathcal{R}_o^\vee = \mathsf{F}$ of $\mathfrak{h}_o^\vee = \mathfrak{sp}(3)$. For $i = 1, 2$, we have $\mathcal{R}_{o,\mathcal{P}_i}^\vee = \mathsf{F}_i$ of $\mathfrak{f}_{\mathcal{P}_i} = \mathfrak{sp}(3)_i$. From this we compute that $Y_F' = 0$.

In this example, we have multiple kinds of twist lines, so the global contribution is provided by the representation (23) which becomes

$$(\Lambda^2)_1 \otimes (\Lambda^2)_2, \tag{59}$$

where $(\Lambda_2)_i$ for $i = 1, 2$ is the 2-index antisymmetric irrep for $\mathfrak{f}_{\mathcal{P}_i} = \mathfrak{sp}(3)_i$. Thus $\widetilde{Y}_F$ is also trivial, and hence the full $Y_F = 0$. Thus, the manifest flavor symmetry group is

$$\mathcal{F} = PSp(3)_1 \times PSp(3)_2, \tag{60}$$

where $PSp(3)_i = Sp(3)_i / \mathbb{Z}_2$.

We can confirm this result using the analysis of Section 2.2 since the resulting $4d$ $\mathcal{N} = 2$ theory admits a duality frame with the following Lagrangian description [19]

$$\left[\mathfrak{sp}(3)_1\right] \xrightarrow[\phantom{xxx}]{\frac{1}{2}\mathsf{F}} \overset{S}{\phantom{x}} \mathfrak{so}(8) \xrightarrow[\phantom{xxx}]{\mathsf{F}} \overset{\frac{1}{2}\mathsf{F}}{\phantom{x}} \left[\mathfrak{sp}(3)_2\right]. \tag{61}$$

Clearly the $\mathbb{Z}_2$ center of each $Sp(3)$ is gauged by a $\mathbb{Z}_2$ factor in the center of $Spin(8)$ gauge group. This leads us to the flavor symmetry group displayed in (60).

### 4.3 Interacting SCFT Fixtures

$\underline{T_N \text{ and } E_6 \text{ MN Theories}}$ : Consider compactifying $\mathfrak{g} = \mathfrak{su}(n)$ $6d$ $\mathcal{N} = (2, 0)$ theory on a sphere with three untwisted regular maximal punctures $\mathcal{P}_i$. Each puncture has $\mathfrak{f}_{\mathcal{P}_i} = \mathfrak{su}(n)_i$. The representation $\mathcal{R}_o^\vee$ is the fundamental representation of $\mathfrak{h}_o^\vee = \mathfrak{g} = \mathfrak{su}(n)$ and thus $\mathcal{R}_{o,\mathcal{P}_i}^\vee$ is the fundamental representation $\mathsf{F}_i$ of $\mathfrak{f}_{\mathcal{P}_i} = \mathfrak{su}(n)_i$. From this we compute that $Y_F' = 0$.

The representation (20) becomes

$$\mathsf{F}_1 \otimes \mathsf{F}_2 \otimes \mathsf{F}_3, \tag{62}$$

implying that $\widetilde{Y}_F$ is generated by the element $(1, 1, 1) \in \widehat{Z}_{F,\mathcal{P}_1} \oplus \widehat{Z}_{F,\mathcal{P}_2} \oplus \widehat{Z}_{F,\mathcal{P}_3} \simeq (\mathbb{Z}/n\mathbb{Z})_1 \oplus (\mathbb{Z}/n\mathbb{Z})_2 \oplus (\mathbb{Z}/n\mathbb{Z})_3$. From this we compute that the manifest flavor symmetry group is

$$\mathcal{F} = \frac{SU(n)_1 \times SU(n)_2 \times SU(n)_3}{\mathbb{Z}_n^{1,2} \times \mathbb{Z}_n^{2,3}}, \tag{63}$$

where $\mathbb{Z}_n^{1,2}$ is the $\mathbb{Z}_n$ subgroup generated by the element $\left(\frac{1}{n}, -\frac{1}{n}, 0\right)$ of the $(\mathbb{Z}_n)_1 \times (\mathbb{Z}_n)_2 \times (\mathbb{Z}_n)_3$ center. The $\mathbb{Z}_n^{2,3}$ factor is generated by the element $\left(0, \frac{1}{n}, -\frac{1}{n}\right)$.

For $n \geq 4$, the manifest flavor symmetry algebra $\mathfrak{f} = \mathfrak{f}_{\mathcal{P}_1} \oplus \mathfrak{f}_{\mathcal{P}_2} \oplus \mathfrak{f}_{\mathcal{P}_3} = \mathfrak{su}(n)_1 \oplus \mathfrak{su}(n)_2 \oplus \mathfrak{su}(n)_3$ is true flavor symmetry algebra. Consequently, the manifest flavor symmetry group (63) is the true flavor symmetry group.

On the other hand, for $n = 3$, we obtain the $E_6$ Minahan-Nemeschansky (MN) theory [20] whose true flavor symmetry algebra is $\mathfrak{f}_{\text{true}} = \mathfrak{e}_6$. In this case, following the analysis of Section 2.3, let us try to determine possible global forms $\mathcal{F}_{\text{true}}$ of $\mathfrak{f}_{\text{true}}$ with the constraint that the global form corresponding to manifest flavor symmetry $\mathfrak{f} = \mathfrak{su}(3)^3$ is given by (63) for $n = 3$. We have two possibilities to consider: $E_6$ and $E_6/\mathbb{Z}_3$. Let us decompose the representations of $\mathfrak{e}_6$ in terms of the $\mathfrak{su}(3)^3$ subalgebra:

- The adjoint $\mathsf{A} = \mathbf{78}$ of $\mathfrak{e}_6$ breaks as $(\mathsf{A}, \mathbf{1}, \mathbf{1}) \oplus (\mathbf{1}, \mathsf{A}, \mathbf{1}) \oplus (\mathbf{1}, \mathbf{1}, \mathsf{A}) \oplus (\mathsf{F}, \mathsf{F}, \mathsf{F}) \oplus (\bar{\mathsf{F}}, \bar{\mathsf{F}}, \bar{\mathsf{F}})$ of $\mathfrak{su}(3)^3$. This generates a sub-lattice $Y_F$ which is the same as $\widetilde{Y}_F$ (for $n = 3$) discussed above. Thus, $\mathcal{F}_{\text{true}} = E_6/\mathbb{Z}_3$ would lead to manifest $\mathcal{F}$ being the one provided by (63) for $n = 3$.

- On the other hand, the fundamental representation $\mathsf{F} = \mathbf{27}$ of $\mathfrak{e}_6$ breaks as $(\bar{\mathsf{F}}, \mathbf{1}, \mathsf{F}) \oplus (\mathsf{F}, \bar{\mathsf{F}}, \mathbf{1}) \oplus (\mathbf{1}, \mathsf{F}, \bar{\mathsf{F}})$ of $\mathfrak{su}(3)^3$. The lattice $Y_F$ corresponding to it is generated by the elements $(1, -1, 0)$ and $(0, 1, -1)$ in $\widehat{Z}_F = (\mathbb{Z}/3\mathbb{Z})_1 \oplus (\mathbb{Z}/3\mathbb{Z})_2 \oplus (\mathbb{Z}/3\mathbb{Z})_3$. Thus, $\mathcal{F}_{\text{true}} = E_6$ would lead to $\mathcal{F} = SU(3)^3/\mathbb{Z}_3$ where $\mathbb{Z}_3$ is the diagonal of the $\mathbb{Z}_3^3$ center of $SU(3)^3$. This manifest $\mathcal{F}$ is different from the manifest $\mathcal{F}$ appearing in (63) for $n = 3$.

Thus, the knowledge of the manifest flavor group (63) leads us to conclude that the true flavor group for $E_6$ MN theory is

$$\mathcal{F}_{\text{true}} = \frac{E_6}{\mathbb{Z}_3}. \tag{64}$$

We can also confirm this result by studying the representations of $\mathfrak{e}_6$ appearing in the superconformal index of $E_6$ MN theory. From the result of [21], we find irreps of dimension $\mathbf{78}$ and $\mathbf{650}$, where the latter irrep is found in the irrep decomposition of the representation $\mathsf{F} \otimes \bar{\mathsf{F}}$ of $\mathfrak{e}_6$. Both of these representations carry trivial charge under the $\mathbb{Z}_3$ center of $E_6$. Thus, the proposed true flavor symmetry group (64) is consistent with the superconformal index appearing in [21]. In a similar fashion, we see that the index of $T_n$ theory presented in [22] is comprised of the adjoint representations and the representations of the form $R \otimes R \otimes R$ of $\mathfrak{f} = \mathfrak{su}(n)^3$ such that $R$ is an irrep of $\mathfrak{su}(n)$, which verifies our result (63).

**$E_7$ MN Theory :**

$$\mathfrak{g} = \mathfrak{su}(4)$$

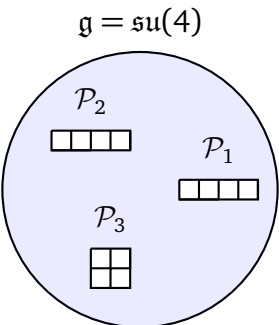

We have $\mathfrak{f}_{\mathcal{P}_i} = \mathfrak{su}(4)_i$ for $i = 1, 2$ and $\mathfrak{f}_{\mathcal{P}_3} = \mathfrak{su}(2)$. The representation $\mathcal{R}_o^\vee$ is the fundamental representation of $\mathfrak{h}_o^\vee = \mathfrak{g} = \mathfrak{su}(4)$. We have $\mathcal{R}_{o,\mathcal{P}_i}^\vee$ is the fundamental representation $\mathsf{F}_i$ of $\mathfrak{f}_{\mathcal{P}_i} = \mathfrak{su}(4)_i$ and $\mathcal{R}_{o,\mathcal{P}_3}^\vee = 2 \cdot \mathsf{F}$ of $\mathfrak{f}_{\mathcal{P}_3} = \mathfrak{su}(2)$. From this we compute that $Y_F' = 0$.

The representation (20) becomes

$$\mathsf{F}_1 \otimes \mathsf{F}_2 \otimes (2 \cdot \mathsf{F}), \tag{65}$$

implying that $\widetilde{Y}_F$ is generated by the element $(1, 1, 1) \in \widehat{Z}_{F,\mathcal{P}_1} \oplus \widehat{Z}_{F,\mathcal{P}_2} \oplus \widehat{Z}_{F,\mathcal{P}_3} \simeq (\mathbb{Z}/4\mathbb{Z})_1 \oplus (\mathbb{Z}/4\mathbb{Z})_2 \oplus \mathbb{Z}/2\mathbb{Z}$. From this we compute that the manifest flavor symmetry group is

$$\mathcal{F} = \frac{SU(4)_1 \times SU(4)_2 \times SU(2)}{\mathbb{Z}_4^{1,2} \times \mathbb{Z}_2^{2,3}}, \tag{66}$$

where $\mathbb{Z}_4^{1,2}$ is the $\mathbb{Z}_4$ subgroup generated by the element $\left(\frac{1}{4}, -\frac{1}{4}, 0\right)$ of the $(\mathbb{Z}_4)_1 \times (\mathbb{Z}_4)_2 \times \mathbb{Z}_2$ center. The $\mathbb{Z}_2^{2,3}$ factor is generated by the element $\left(0, \frac{1}{2}, \frac{1}{2}\right)$.

This theory is the $E_7$ MN theory whose true flavor symmetry algebra is $\mathfrak{f}_{\text{true}} = \mathfrak{e}_7$. In this case, following the analysis of Section 2.3, let us try to determine possible global forms $\mathcal{F}_{\text{true}}$

of $\mathfrak{f}_{\text{true}}$ with the constraint that the global form corresponding to manifest flavor symmetry $\mathfrak{f} = \mathfrak{su}(4)^2 \oplus \mathfrak{su}(2)$ is given by (66). We have two possibilities to consider: $E_7$ and $E_7/\mathbb{Z}_2$. Let us decompose the representations of $\mathfrak{e}_7$ in terms of the $\mathfrak{su}(4)^2 \oplus \mathfrak{su}(2)$ subalgebra:

- The adjoint $\mathsf{A} = \mathbf{133}$ of $\mathfrak{e}_7$ breaks as $(\mathsf{A}, \mathbf{1}, \mathbf{1}) \oplus (\mathbf{1}, \mathsf{A}, \mathbf{1}) \oplus (\mathbf{1}, \mathbf{1}, \mathsf{A}) \oplus (\mathsf{F}, \mathsf{F}, \mathsf{F}) \oplus (\bar{\mathsf{F}}, \bar{\mathsf{F}}, \mathsf{F}) \oplus (\Lambda^2, \Lambda^2, \mathbf{1})$ of $\mathfrak{su}(4)^2 \oplus \mathfrak{su}(2)$. This generates a sub-lattice $Y_F$ which is the same as $\widetilde{Y}_F$ discussed above. Thus, $\mathcal{F}_{\text{true}} = E_7/\mathbb{Z}_2$ would lead to manifest $\mathcal{F}$ being the one appearing in (66).

- On the other hand, the fundamental representation $\mathsf{F} = \mathbf{56}$ of $\mathfrak{e}_7$ breaks as $(\Lambda^2, \mathbf{1}, \mathsf{F}) \oplus (\mathsf{F}, \bar{\mathsf{F}}, \mathbf{1}) \oplus (\mathbf{1}, \Lambda^2, \mathsf{F}) \oplus (\bar{\mathsf{F}}, \mathsf{F}, \mathbf{1})$ of $\mathfrak{su}(4)^2 \oplus \mathfrak{su}(2)$. The lattice $Y_F$ corresponding to it is generated by the elements $(1, -1, 0)$ and $(0, 2, 1)$ in $\widehat{Z}_F = (\mathbb{Z}/4\mathbb{Z})_1 \oplus (\mathbb{Z}/4\mathbb{Z})_2 \oplus \mathbb{Z}/2\mathbb{Z}$. Thus, $\mathcal{F}_{\text{true}} = E_7$ would lead to $\mathcal{F} = \frac{SU(4)^2 \times SU(2)}{\mathbb{Z}_4}$ where $\mathbb{Z}_4$ is generated by the element $\left(\frac{1}{4}, \frac{1}{4}, \frac{1}{2}\right)$. This manifest $\mathcal{F}$ is different from the manifest $\mathcal{F}$ appearing in (66).

Thus, the knowledge of the manifest flavor group (66) leads us to conclude that the true flavor group for $E_7$ MN theory is

$$\mathcal{F}_{\text{true}} = \frac{E_7}{\mathbb{Z}_2}. \tag{67}$$

We can also confirm this result by studying the representations of $\mathfrak{e}_7$ appearing in the index of $E_7$ MN theory studied in [22–24].

**Rank-1 $SU(4)_{14}$ SCFT – $\mathbb{Z}_3$ twisted $D_4$ example :** Consider compactifying $\mathfrak{g} = \mathfrak{so}(8)$ 6d $\mathcal{N} = (2,0)$ theory on a sphere with three regular punctures: one $\mathbb{Z}_3$ twisted emitting a $\mathbb{Z}_3$ twist line, carrying BC label $A_1$, and labeled $\mathcal{P}_1$; one $\mathbb{Z}_3$ twisted absorbing the $\mathbb{Z}_3$ twist line, carrying BC label $A_1$, and labeled $\mathcal{P}_2$; and one untwisted specified by the Nahm YD which is the *transpose of* ⊞⊞⊞⊞⊞ labeled $\mathcal{P}_3$. We have $\mathfrak{f}_{\mathcal{P}_i} = \mathfrak{su}(2)_i$ for $i = 1, 2$ and $\mathfrak{f}_{\mathcal{P}_3}$ is trivial. The representation $\mathcal{R}_o^\vee = \mathbf{7}$ of $\mathfrak{h}_o^\vee = \mathfrak{g}_2$ which reduces to $\mathcal{R}_{o,\mathcal{P}_i}^\vee = \mathsf{A} \oplus 2 \cdot \mathsf{F}$ of $\mathfrak{f}_{\mathcal{P}_i} = \mathfrak{su}(2)_i$ for $i = 1, 2$. From this we compute that $Y_F' = \widehat{Z}_F = \widehat{Z}_{F,\mathcal{P}_1} \oplus \widehat{Z}_{F,\mathcal{P}_2} \simeq (\mathbb{Z}/2\mathbb{Z})_1 \oplus (\mathbb{Z}/2\mathbb{Z})_2$.

Thus full $Y_F = \widehat{Z}_F$ implying that the manifest flavor symmetry group is

$$\mathcal{F} = SU(2)_1 \times SU(2)_2. \tag{68}$$

The true flavor symmetry algebra of this theory is $\mathfrak{f}_{\text{true}} = \mathfrak{su}(4)$ such that the adjoint rep of $\mathfrak{su}(4)$ breaks as $(\mathsf{A}, \mathbf{1}) \oplus (\mathbf{1}, \mathsf{A}) \oplus (\mathsf{F}, \mathsf{F}) \oplus (\mathbf{1}, \mathbf{1})$ under $\mathfrak{f} = \mathfrak{su}(2)^2$ subalgebra. There global form (68) of the manifest flavor symmetry implies that the global form of the true flavor symmetry must be

$$\mathcal{F}_{\text{true}} = SU(4). \tag{69}$$

This result can be confirmed using the index for this theory computed in [16].

**$\widetilde{T}_3$ theory :**

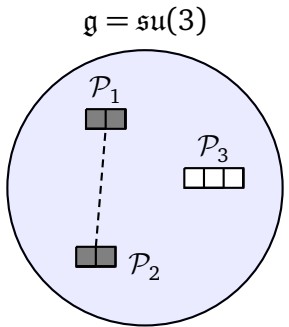

We have $\mathfrak{f}_{\mathcal{P}_i} = \mathfrak{su}(2)_i$ for $i = 1, 2$ and $\mathfrak{f}_{\mathcal{P}_3} = \mathfrak{su}(3)$. For $\mathcal{P}_i$ $i = 1, 2$ we have $\mathfrak{h}_o^\vee = \mathfrak{su}(2)$ with corresponding $\mathcal{R}_o^\vee = \mathsf{F}$ of $\mathfrak{h}_o^\vee = \mathfrak{su}(2)$ and $\mathcal{R}_o^\vee = \mathsf{F}_i$ of $\mathfrak{f}_{\mathcal{P}_i} = \mathfrak{su}(2)_i$. For $\mathcal{P}_3$, we have $\mathfrak{h}_o^\vee = \mathfrak{g} = \mathfrak{su}(3)$ with associated $\mathcal{R}_o^\vee = \mathsf{F}$ of $\mathfrak{h}_o^\vee = \mathfrak{g} = \mathfrak{su}(3)$ and $\mathcal{R}_{o,\mathcal{P}_3}^\vee = \mathsf{F}$ of $\mathfrak{f}_{\mathcal{P}_3} = \mathfrak{su}(3)$. From this we compute that $Y_F' = 0$.

All twist lines are of the same type and they are associated to $\mathfrak{h}_o^\vee = \mathfrak{su}(2)$ which has corresponding $\mathcal{R}_o^\vee = \mathsf{F}$ of $\mathfrak{h}_o^\vee = \mathfrak{su}(2)$ and $\mathsf{R}_o = \mathsf{A}$ of $\mathfrak{g} = \mathfrak{su}(3)$. The representation (22) becomes $\mathcal{R} = \mathsf{F}_1 \otimes \mathsf{F}_2 \otimes \mathsf{A}$ of $\mathfrak{f}_{\mathcal{P}_1} \oplus \mathfrak{f}_{\mathcal{P}_2} \oplus \mathfrak{f}_{\mathcal{P}_3} = \mathfrak{su}(2)_1 \oplus \mathfrak{su}(2)_2 \oplus \mathfrak{su}(3)$. From this, we conclude that the full $Y_F$ is generated by the element $(1, 1, 0) \in \widehat{Z}_{F,\mathcal{P}_1} \times \widehat{Z}_{F,\mathcal{P}_2} \times \widehat{Z}_{F,\mathcal{P}_3} \simeq (\mathbb{Z}/2\mathbb{Z})_1 \times (\mathbb{Z}/2\mathbb{Z})_2 \times \mathbb{Z}/3\mathbb{Z}$ which implies that the manifest flavor group is

$$\mathcal{F} = \frac{SU(2)_1 \times SU(2)_2}{\mathbb{Z}_2^{1,2}} \times PSU(3) = SO(4) \times PSU(3), \tag{70}$$

where $\mathbb{Z}_2^{1,2}$ is the diagonal $\mathbb{Z}_2$ of the $(\mathbb{Z}_2)_1 \times (\mathbb{Z}_2)_2$ center of $SU(2)_1 \times SU(2)_2$. This result matches with the index for this theory appearing in [4].

**Distinguishing** $4d$ $\mathcal{N} = 2$ **SCFTs :** Now we consider a pair of interacting $4d$ $\mathcal{N} = 2$ SCFTs with very similar Class S constructions. These SCFTs have the same set of invariants, but were predicted to be distinguished by subtle differences in the global forms of flavor groups of these two SCFTs [5]. We will only consider the first pair, namely "Theory I", appearing in [5], but the other pairs can also be discussed in the same fashion. We will find complete agreement with the flavor symmetry groups proposed in [5] by studying the Schur index.

The first theory in the pair is described by

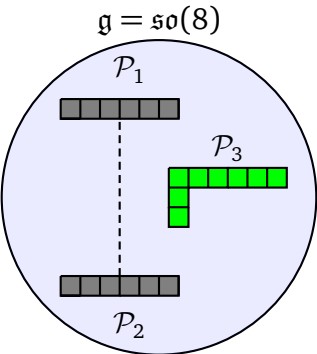

We have $\mathfrak{f}_{\mathcal{P}_i} = \mathfrak{sp}(3)_i$ for $i = 1, 2$ and $\mathfrak{f}_{\mathcal{P}_3} = \mathfrak{sp}(2)$. For $\mathcal{P}_i$ $i = 1, 2$ we have $\mathfrak{h}_o^\vee = \mathfrak{sp}(3)$ with corresponding $\mathcal{R}_o^\vee = \mathsf{F}$ of $\mathfrak{h}_o^\vee = \mathfrak{sp}(3)$ and $\mathcal{R}_o^\vee = \mathsf{F}_i$ of $\mathfrak{f}_{\mathcal{P}_i} = \mathfrak{sp}(3)_i$. For $\mathcal{P}_3$, we have $\mathfrak{h}_o^\vee = \mathfrak{g} = \mathfrak{so}(8)$ with associated $\mathcal{R}_{o,s}^\vee = \mathsf{S}$ and $\mathcal{R}_{o,c}^\vee = \mathsf{C}$ of $\mathfrak{h}_o^\vee = \mathfrak{g} = \mathfrak{so}(8)$, and $\mathcal{R}_{o,s,\mathcal{P}_3}^\vee = 2 \cdot \mathsf{F}$ and $\mathcal{R}_{o,c,\mathcal{P}_3}^\vee = 2 \cdot \mathsf{F}$ of $\mathfrak{f}_{\mathcal{P}_3} = \mathfrak{sp}(2)$. From this we compute that $Y_F' = 0$.

All twist lines are of the same type and they are associated to $\mathfrak{h}_o^\vee = \mathfrak{sp}(3)$ which has corresponding $\mathcal{R}_o^\vee = \mathsf{F}$ of $\mathfrak{h}_o^\vee = \mathfrak{sp}(3)$ and $\mathsf{R}_o = \mathsf{A}$ of $\mathfrak{g} = \mathfrak{so}(8)$. The representation (22) becomes $\mathcal{R} = \mathsf{F}_1 \otimes \mathsf{F}_2 \otimes (3 \cdot \mathbf{1} \oplus \Lambda^2)$ of $\mathfrak{f}_{\mathcal{P}_1} \oplus \mathfrak{f}_{\mathcal{P}_2} \oplus \mathfrak{f}_{\mathcal{P}_3} = \mathfrak{sp}(3)_1 \oplus \mathfrak{sp}(3)_2 \oplus \mathfrak{sp}(2)$. From this, we conclude that the full $Y_F$ is generated by the element $(1, 1, 0) \in \widehat{Z}_{F,\mathcal{P}_1} \times \widehat{Z}_{F,\mathcal{P}_2} \times \widehat{Z}_{F,\mathcal{P}_3} \simeq (\mathbb{Z}/2\mathbb{Z})_1 \times (\mathbb{Z}/2\mathbb{Z})_2 \times \mathbb{Z}/2\mathbb{Z}$ which implies that the manifest flavor group is

$$\mathcal{F} = \frac{Sp(3)_1 \times Sp(3)_2}{\mathbb{Z}_2} \times SO(5), \tag{71}$$

where $\mathbb{Z}_2$ appearing in the denominator is the diagonal $\mathbb{Z}_2$ of the $(\mathbb{Z}_2)_1 \times (\mathbb{Z}_2)_2$ center of $Sp(3)_1 \times Sp(3)_2$, and $SO(5)$ is the non-simply-connected group $Sp(2)/\mathbb{Z}_2$. This result matches with the prediction of [5].

The second theory in the pair is described by

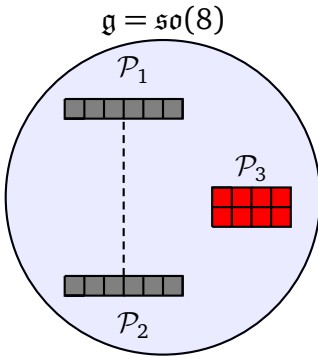

$$\mathfrak{g} = \mathfrak{so}(8)$$

We again have $\mathfrak{f}_{\mathcal{P}_i} = \mathfrak{sp}(3)_i$ for $i = 1, 2$ and $\mathfrak{f}_{\mathcal{P}_3} = \mathfrak{sp}(2)$. For $\mathcal{P}_i$ $i = 1, 2$ we again have $\mathfrak{h}_o^\vee = \mathfrak{sp}(3)$ with corresponding $\mathcal{R}_o^\vee = \mathsf{F}$ of $\mathfrak{h}_o^\vee = \mathfrak{sp}(3)$ and $\mathcal{R}_o^\vee = \mathsf{F}_i$ of $\mathfrak{f}_{\mathcal{P}_i} = \mathfrak{sp}(3)_i$. For $\mathcal{P}_3$, we now have $\mathfrak{h}_o^\vee = \mathfrak{g} = \mathfrak{so}(8)$ with associated $\mathcal{R}_{o,s}^\vee = \mathsf{S}$ and $\mathcal{R}_{o,c}^\vee = \mathsf{C}$ of $\mathfrak{h}_o^\vee = \mathfrak{g} = \mathfrak{so}(8)$, and $\mathcal{R}_{o,s,\mathcal{P}_3}^\vee = 3 \cdot \mathbf{1} \oplus \Lambda^2$ and $\mathcal{R}_{o,c,\mathcal{P}_3}^\vee = 2 \cdot \mathsf{F}$ of $\mathfrak{f}_{\mathcal{P}_3} = \mathfrak{sp}(2)$. From this we compute that $Y_F' = 0$.

All twist lines are again of the same type and they are associated to $\mathfrak{h}_o^\vee = \mathfrak{sp}(3)$ which has corresponding $\mathcal{R}_o^\vee = \mathsf{F}$ of $\mathfrak{h}_o^\vee = \mathfrak{sp}(3)$ and $\mathsf{R}_o = \mathsf{A}$ of $\mathfrak{g} = \mathfrak{so}(8)$. The representation (22) now becomes $\mathcal{R} = \mathsf{F}_1 \otimes \mathsf{F}_2 \otimes (2 \cdot \mathsf{F})$ of $\mathfrak{f}_{\mathcal{P}_1} \oplus \mathfrak{f}_{\mathcal{P}_2} \oplus \mathfrak{f}_{\mathcal{P}_3} = \mathfrak{sp}(3)_1 \oplus \mathfrak{sp}(3)_2 \oplus \mathfrak{sp}(2)$. From this, we conclude that the full $Y_F$ is generated by the element $(1, 1, 1) \in \widehat{Z}_{F,\mathcal{P}_1} \times \widehat{Z}_{F,\mathcal{P}_2} \times \widehat{Z}_{F,\mathcal{P}_3} \simeq (\mathbb{Z}/2\mathbb{Z})_1 \times (\mathbb{Z}/2\mathbb{Z})_2 \times \mathbb{Z}/2\mathbb{Z}$ which implies that the manifest flavor group is

$$\mathcal{F} = \frac{Sp(3)_1 \times Sp(3)_2 \times Sp(2)}{\mathbb{Z}_2^{1,2} \times \mathbb{Z}_2^{2,3}}, \tag{72}$$

where $\mathbb{Z}_2^{1,2}$ is the diagonal $\mathbb{Z}_2$ of the $(\mathbb{Z}_2)_1 \times (\mathbb{Z}_2)_2$ center of $Sp(3)_1 \times Sp(3)_2$, and $\mathbb{Z}_2^{2,3}$ is the diagonal $\mathbb{Z}_2$ of the $(\mathbb{Z}_2)_2 \times \mathbb{Z}_2$ center of $Sp(3)_2 \times Sp(2)$. This result again matches with the prediction of [5].

## Acknowledgements

The author thanks Fabio Apruzzi, Jihwan Oh, Sakura Schäfer-Nameki and Yuji Tachikawa for discussions. This work is supported by ERC grants 682608 and 787185 under the European Union's Horizon 2020 programme.

## A Glossary of Notation

- $\mathfrak{T}$: A quantum field theory.

- $\mathcal{F}$: 0-form symmetry group or flavor symmetry group.

- $\mathfrak{f}$: 0-form symmetry algebra or flavor symmetry algebra.

- $F$: A Lie group with Lie algebra being the flavor algebra $\mathfrak{f}$, which in general is a central extension of the flavor group $\mathcal{F}$.

- $Z_F$: Center of the group $F$.

- $\widehat{Z}_F$: Pontryagin dual of $Z_F$.

- $Y_F$: Subgroup of $\widehat{Z}_F$ capturing the "flavor center charges" of genuine local operators under $Z_F$.

- $\mathcal{Z}$: Subgroup of $Z_F$ under which all genuine local operators have zero charge.

- $\widehat{\mathcal{Z}}$: Pontryagin dual of $\mathcal{Z}$.

- $G$: Gauge group of a gauge theory.

- $Z_G$: Center of the group $G$.

- $\widehat{Z}_G$: Pontryagin dual of $Z_G$.

- $\mathcal{M}$: Subgroup of $\widehat{Z}_G \times \widehat{Z}_F$ capturing the gauge and flavor center charges of matter fields in a gauge theory.

- $\mathcal{E}$: Subgroup of $Z_G \times Z_F$ under which all matter fields have zero charge.

- $\mathfrak{f}_m$: Manifest flavor symmetry algebra of a Class S theory.

- $\mathcal{F}_m$: Manifest flavor symmetry group of a Class S theory.

- $\mathcal{P}_i$: A puncture in a Class S compactification.

- $\mathfrak{g}$: An A, D, E Lie algebra associated to a $6d$ $\mathcal{N} = (2,0)$ theory. Can also be a semi-simple Lie algebra describing gauge algebra of a gauge theory.

- $\mathcal{G}$: Simply connected group associated to the A, D, E Lie algebra $\mathfrak{g}$ associated to a $6d$ $\mathcal{N} = (2,0)$ theory.

- $Z(\mathcal{G})$: Center of $\mathcal{G}$.

- $\widehat{Z}(\mathcal{G})$: Pontryagin dual of $Z(\mathcal{G})$.

- $\mathcal{O}_{\mathfrak{g}}$: Group of outer automorphisms modulo inner automorphisms of the A, D, E Lie algebra $\mathfrak{g}$ associated to a $6d$ $\mathcal{N} = (2,0)$ theory. Acts as 0-form symmetry group of the associated $6d$ $\mathcal{N} = (2,0)$ theory.

- $o$: An element of $\mathcal{O}_{\mathfrak{g}}$.

- $\mathfrak{h}_o$: Subgroup of $\mathfrak{g}$ left invariant by an outer-automorphism lying in the class $o \in \mathcal{O}_{\mathfrak{g}}$. See Table 1 for the possible $\mathfrak{h}_o$.

- $\mathfrak{h}_o^{\vee}$: Langlands dual of $\mathfrak{h}_o$.

- $R_o$: Representation of $\mathfrak{h}_o$ descending from a representation $R$ of $\mathfrak{g}$ left invariant by the action of $o$.

- $R_o^{\vee}$: Representation of $\mathfrak{h}_o^{\vee}$ exchanged with representation $R_o$ of $\mathfrak{h}_o$ under Langlands duality.

- $\mathcal{R}_o^{\vee}$: Specific representation of $\mathfrak{h}_o^{\vee}$. Table 1 specifies the representation for each choice of $o$ and $\mathfrak{g}$.

- $\mathsf{R}_o$: Representation of $\mathfrak{g}$ that descends to representation $\mathcal{R}_o^{\vee}$.

- $\mathfrak{f}_{\mathcal{P}}$: Flavor algebra associated to a puncture $\mathcal{P}$.

- $F_{\mathcal{P}}$: A Lie group with algebra $\mathfrak{f}_{\mathcal{P}}$.

- $Z_{F,\mathcal{P}}$: Center of $F_{\mathcal{P}}$.

- $\widehat{Z}_{F,\mathcal{P}}$: Pontryagin dual of $Z_{F,\mathcal{P}}$.

- $\mathcal{R}^\vee_{o,\mathcal{P}}$: Representation of $\mathfrak{f}_{\mathcal{P}}$ obtained by viewing the representation $\mathcal{R}^\vee_o$ of $\mathfrak{h}^\vee_o$ from the point of view of $\mathfrak{f}_{\mathcal{P}} \subseteq \mathfrak{h}^\vee_o$.

- $Y_{F,\mathcal{P}}$: Subgroup of $\widehat{Z}_{F,\mathcal{P}}$ capturing the charges under $Z_{F,\mathcal{P}}$ of the irreps of $\mathfrak{f}_{\mathcal{P}}$ (and their tensor products) arising in the irrep decomposition of the representation $\mathcal{R}^\vee_{o,\mathcal{P}}$ of $\mathfrak{f}_{\mathcal{P}}$.

- $Y'_F$: Subgroup of $\widehat{Z}_F$ of a Class S theory obtained by taking the direct sum over all punctures $\mathcal{P}$ of groups $Y_{F,\mathcal{P}}$. See equation (15).

- $\mathcal{R}$: Representation of $\mathfrak{f}$ associated to a Class S compactification. See Section 3.3 for details.

- $\widetilde{Y}_F$: Subgroup of $\widehat{Z}_F$ of a Class S theory capturing the charges under $Z_F$ of the irreps of $\mathfrak{f}$ (and their tensor products) arising in the irrep decomposition of the representation $\mathcal{R}$ of $\mathfrak{f}$. The group $Y_F$ of a Class S theory can be expressed in terms of groups $Y'_F$ and $\widetilde{Y}_F$ as in equation (19).

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
