# Peer review of "Global Form of Flavor Symmetry Groups in 4d N=2 Theories of Class S"

_SciPost Physics, doi:SciPost Phys. 12, 183 (2022)_

## Round 2 · Referee Report · Anonymous (Referee 1) · 2022-1-28

Strengths

  1. The results are interesting and useful.
  2. The method discussed in the article can be applied to general theories of class $\mathcal{S}$.

Weaknesses

  1. The presentation of the material and style of the article could be significantly improved so that the paper is more readable.
  2. The choice of notation renders the article heavy and unreadable.
  3. Many computational details need to be to be spelt out explicitly.

Report

This article provides a general method to compute the the global form of flavor symmetry in the theories of class $\mathcal{S}$, with regular punctures and possibly with outer-automorphism twists. The author demonstrated the application of such a method in wide range of examples. Although the results are interesting and useful, the referee would like to ask the following improvement prior to publication.

Requested changes

  1. The authors mentioned the lattices of charges $Y'_F$, $\tilde{Y}_F$ and $Y_F$ in Eqs. (3.6), (3.9) and (3.10), respectively. However, none of the detail is spelt out in the subsequent examples. For example, at the end page 23, how did the authors compute $\tilde{Y}_F$? Since these are important for obtaining the main results in the article, I highly recommend the author to spell out the detail explicitly for non-trivial examples, for example in an appendix.

  2. Given the lattice $Y_F$, how did the author compute the quotient $\mathcal{Z}$ and its generator? (For example, in the sentence below Eq. (4.6).) Again, the author definitely should elucidate the detail explicitly.

  3. Could the author explain why the complex conjugate representation disappears from Eq. (3.2) to Eq. (3.11)?

---

## Round 2 · Referee Report · Anonymous (Referee 2) · 2022-3-31

Strengths

  1. The article has useful results.
  2. The author discusses several examples including the cases of surfaces with twists.
  3. The author discusses how the global form of the full symmetry group can often be obtained from the global form of the manifest symmetry.
  4. The author also discusses an application of his results on the topic of distinguishing between distinct SCFTs.

Weaknesses

  1. Notation used in the article is heavy. It could be useful to have a glossary of notation in an appendix so that a reader can look up the meaning of each symbol.

Report

It has become clear over the past decade that the information contained in the global form of the global symmetry group is essential to understanding a QFT better. For example, it helps determine the 't Hooft anomalies of the QFT, which are invariant under an RG flow. For Lagrangian theories, determining the global form is easy; however, for non-Lagrangian theories (for example, those in class S), it was not clear how to answer this question. The author has given a neat answer to this question: it is determined by thinking carefully about the punctures and the surface defects of 6d (2,0) theory. This article adds useful information to the knowledge of the community, and I recommend it for publication.

Requested changes

I suggest that the author includes a glossary of notation in the appendix. This would make the article easier to read.

---

## Round 3 · Referee Report · Anonymous (Referee 1) · 2022-4-21

Report

In the new version of the preprint, the author has addressed the comments of the referees satisfactorily. The referee recommends the article for publication.

---

## Round 3 · List of Changes

The following changes were made after referees' suggestions:

  1. A glossary of notation is included as an appendix.
  2. More details on computation were provided in the example labeled 'Bifundamental Hyper' in Section 4.1. The author hopes that this would act as a good illustration for carrying out similar computations in the rest of Section 4.
  3. An explanation was added after equation (3.7) justifying why equation (3.7) does not have complex conjugates, but equation (3.4) does.

---

## Editorial Decision

published